# Remote Sensing Analysis of Ecological Maintenance in Subtropical Coastal Mountain Area, China

Run Han [1], Jinming Sha [1,2], Xiaomei Li [3,*], Shuhui Lai [1,2], Zejing Lin [1,2], Qixin Lin [1,2] and Jinliang Wang [4]

1   School of Geographical Science, Fujian Normal University, Fuzhou 350007, China; 109092019097@student.fjnu.edu.cn (R.H.); jmsha@fjnu.edu.cn (J.S.); qsx20201099@student.fjnu.edu.cn (S.L.); qsx20211160@student.fjnu.edu (Z.L.); qsx20190896@student.fjnu.edu.cn (Q.L.)
2   Fujian Provincial Key Laboratory of Subtropical Resources and Environment, Fujian Normal University, Fuzhou 350007, China
3   College of Environment Science & Engineering, Fujian Normal University, Fuzhou 350007, China
4   Faculty of Geography, Yunnan Normal University, Kunming 650500, China; jlwang@ynnu.edu.cn
*   Correspondence: lixiaomei@fjnu.edu.cn

**Abstract:** Mountain areas in China account for 69% of the total land area; however, it is still an urgent that we grasp the special ecological structure of mountain areas and maximize the resource advantages of mountain areas under the principle of maintaining a certain ecological level. In this paper, Landsat 5, Landsat 8 and Sentinel-2A images were used as data sources to monitor and analyze land development and ecological change in Gui 'an in 2010, 2013, 2016, 2019 and 2021, so as to explore ecological maintenance mechanisms. Firstly, random forest classification based on multi-source remote sensing data was used to classify land, and the five phases of land-use change were assessed using quantitative analysis, in order to analyze the mountain region's land-use-change characteristics at different stages of development. The results show that Gui 'an has the "two-stage", rapid-development, rapid-recovery mode. Each stage includes a development-and-expansion period and a construction-and-protection period. In the construction period, ecological recovery construction will be emphasized, and the change intensity and rate of the second stage are lower than that of the first stage. Secondly, using a remote sensing ecological index, vegetation coverage, and a landscape index, an ecological evaluation model of the study area was constructed to analyze changes in the ecological environment and its protection in the process of land development. The ecological maintenance status of the five stages was quantitatively monitored using the analysis methods of difference change and coefficient of variation. The results showed that in the first stage of land development and expansion, the landscape pattern and ecological quality fluctuated greatly, and the proportion of ecological quality of an excellent grade decreased by 28.46%. However, in the second stage, the change slowed down and remained unchanged, and gradually moved to the middle and upper level. The results show that there is a close relationship between ecological maintenance and the land development mode, and new mountain towns can maintain ecological quality and achieve sustainable development through a reasonable land development mode. At the end of this paper, the factors affecting the ecological maintenance capacity of Gui 'an are discussed, providing effective reference material and development models for the development of mountainous areas.

**Keywords:** mountain development; ecological maintenance; remote sensing monitoring

## 1. Introduction

In the early 1970s, the concept of ecosystem services was formally proposed, and gradually became known and applied under the research and promotion of HOLDREN et al. [1] and EHRLICH et al. [2]. Now, the quantitative evaluation of ecosystem service function using models such as ecosystem services and trade-offs [3], the ARIS model and the CITY green model are developed using spatial technology with remote sensing (RS) data. Land-use/land-cover change (LUCC) can directly change the function and structure of an

ecosystem, and is monitored extensively and accurately with the RS data [4]. With regard to a mountain ecological-environment quality, with RS data, ecological parameters such as plant diversity, primary productivity, leaf area index, water resources and LUCC can easily be directly monitored; animal diversity and human social and economic activities also can be indirectly monitored [5–7]. Additionally, by utilizing the advantages of multi-source and multi-temporal RS data, the improvement in the temporal and spatial continuity of RS data meets the requirements for mountain ecological-environment monitoring [8].

Since the release of the "Millennium Ecosystem Assessment report" by the United Nations in 2005, the trade-offs and synergies among ecosystem services have been highlighted in the research field. Sustainability and ecological restoration are widely discussed by scholars [9–12]. For the global ecological problems, in the "Convention on Biological Diversity", the United Nations proposed the strategic target of restoring 15% of degraded ecosystems by 2020 [13]. With the fast urbanization in China in subtropical coastal areas, the demand for land is increasing, and the scarcity of land resources is becoming increasingly prominent. Developing hilly areas for building mountain towns has become an important strategy for coastal areas in southern China [14]. However, the ecological environment in mountain areas is more sensitive and fragile. Figuring out how to develop and construct mountain towns scientifically and rationally, under the premise of maintaining a certain level of ecological quality, is seriously important. Normally, urban development leads to a decline in ecological quality and even ecological deterioration. In the interest of the balance and coordination of ecological services, ecological restoration is necessary, although it often fails to fully restore the ecosystem to its original state [15].

This paper puts forward the concept of ecological maintenance for the development of coastal mountain areas. Ecological maintenance is defined as the stability of the eco-environmental quality, and is represented with indicators such as remote sensing ecological index (*RSEI*) and vegetation coverage (*FVC*). It can be quantitatively caculated with the coefficient of variation and difference value of *RSEI* and *FVC*. Ecological maintenance reflects the changing features of eco-environmental quality in new mountain towns from the initial stage of development to the stage of stable development. It aims to keep ecological quality at a certain level during construction, and is conducive to keeping restoration close to the original level to maintaining a natural ecosystem.

Ecological maintenance is influenced by factors such as regional development planning, environmental governance, ecological protection, local inhabitants' lifestyles, and the synergism between resource exploitation and economic development. Due to the complexity of terrain, landform and geological conditions, mountainous areas are ecologically sensitive and vulnerable [16]. Therefore, ecological maintenance is seriously valued for the sustainable development of mountain areas.

This paper takes the subtropic coastal mountain area of Gui 'an in Lianjiang County, Fuzhou, China as the research area. The exploitation of Gui 'an comprises mountain sustainable development with ecological protection, poverty alleviation, leisure tourism, elderly care, a conference center, an IT software park, etc. Therefore, in order to explore how to achieve ecological maintenance in the process of the development and construction of new towns, based on the accurate analysis of land development mode from 2010 to 2021, three evaluation indexes—namely, remote sensing ecological index, vegetation coverage index and landscape pattern index—are used in this paper. The ecological quality change, vegetation cover change and landscape pattern change of Gui 'an are analyzed by means of difference monitoring and coefficient of variation analysis. The relationship between land development and construction characteristics is summarized and the ecological maintenance mechanism is comprehensively analyzed. The paper provides solutions and references for how to realize ecological maintenance in the process of development and construction.

## 2. Materials and Methods

### 2.1. Survey and Data Sources of Study Areas

#### 2.1.1. Study Area

The Gui ′an research area includes the Gui ′an area and Renshan area; the Gui ′an New World is located in the Renshan area, and the Gui ′an hot spring tourism resort is located in the Gui ′an area (Figure 1). Gui ′an is located at 119°20′30″–119°23′40″ east longitude and 26°10′–26°12′30″ north latitude (Figure 1). It is located in Lianjiang County in the north of Fuzhou city, Fujian province, adjacent to Lianjiang County, with an area of about 20 km². Gui′ an has a profound cultural heritage, rich original ecological landscape resources, a beautiful landscape environment, and fresh and comfortable air, and is reputed to be a "natural big oxygen bar in Fuzhou backyard garden". The research area was close to the Fuzhou Ring Expressway, which provides convenient and fast transportation, a unique geographical location, and great development potential. The terrain is mainly composed of mountains, hills, mountain basin and valley, and is surrounded by low hills and hills, so initial development is difficult (Figure 1). Gui ′an has rich geothermal resources. With the implementation of Fuzhou's strategy to build "The Capital of Hot springs in China", Gui ′an hot springs, as a key support project, ushered in rapid development. Gui ′an was just a wilderness before, and the local government was not successful in attracting investment. It believed that Gui ′an area had high mountains and steep roads and was far away from the main city, so it could only rely on hot-spring resources to support some low-density residential buildings. However, under the leadership of the Century Jinyuan Group, the region has been developed, and has successfully enabled the region to realize social self-starting, gathering social wealth, forming an industrial society, enabling local farmers to become waiters and immigrants to help the poor, and realizing local poverty alleviation. In 2019, due to Gui ′an's driving effect on the local economy, Pandu Township successfully removed its status of township and was transformed into a town. In terms of ecological maintenance, it achieved a rapid change in terrain—and fast ecological recovery. It is a model of land development, sustainable development, ecological protection and poverty-alleviation in mountain areas. The Century Jinyuan Group led the enterprise to start the development process of Gui ′an on its own. Since the development and construction, Gui ′an has actively practiced the concept of "clear water and green mountains are gold and silver mountains", in accordance with "343"; this includes the need to: actively integrate into the construction of new Fujian; implement the rural revitalization strategy; adhere to high-quality development to catch up with the three overall actions; and strive to create a new era of "three high" forestry with a high level of forestry ecology, a high-quality forestry industry, and the high well-being of forest people. It has successfully become a model area for mountain development that effectively maintains its ecological level in the process of mountain development.

#### 2.1.2. Data Sources and Pre-Processing

In this paper, medium- and high-resolution remote sensing data were selected, and Sentinel-2A was taken as the main data source. Landsat series data were used to supplement the sentinel-2A in historical image data to form continuous time series data. Therefore, the study data were mainly Landsat 5, Landsat 8, and Sentinel-2A remote sensing images, and the auxiliary data included DEM data, high-resolution Google image data, meteorological data, and relevant planning documents on the official government website. The main data information is shown in Table 1. Landsat 5 in 2010, Landsat 8 in 2013, and Sentinel-2A image data and DEM data in 2016, 2019, and 2021, were obtained through the Geospatial Data Cloud and ESA, respectively. The selected Sentinel data were cloud-free and of good quality (cloud cover ≤ 1%), and the Sentinel-2A images were atmospherically corrected to L2A-level products using the Sen2cor plug-in provided by ESA. Six bands at 20 m resolution were resampled to 10 m resolution using the SNAP software, and a total of 10 bands at 10 m resolution were obtained by combining the four original 10 m resolution bands. The original Landsat5 and Landsat8 image data were preprocessed using ENVI software for

radiometric calibration, atmospheric correction and image clipping. The Google images were used for the selection of classification training samples.

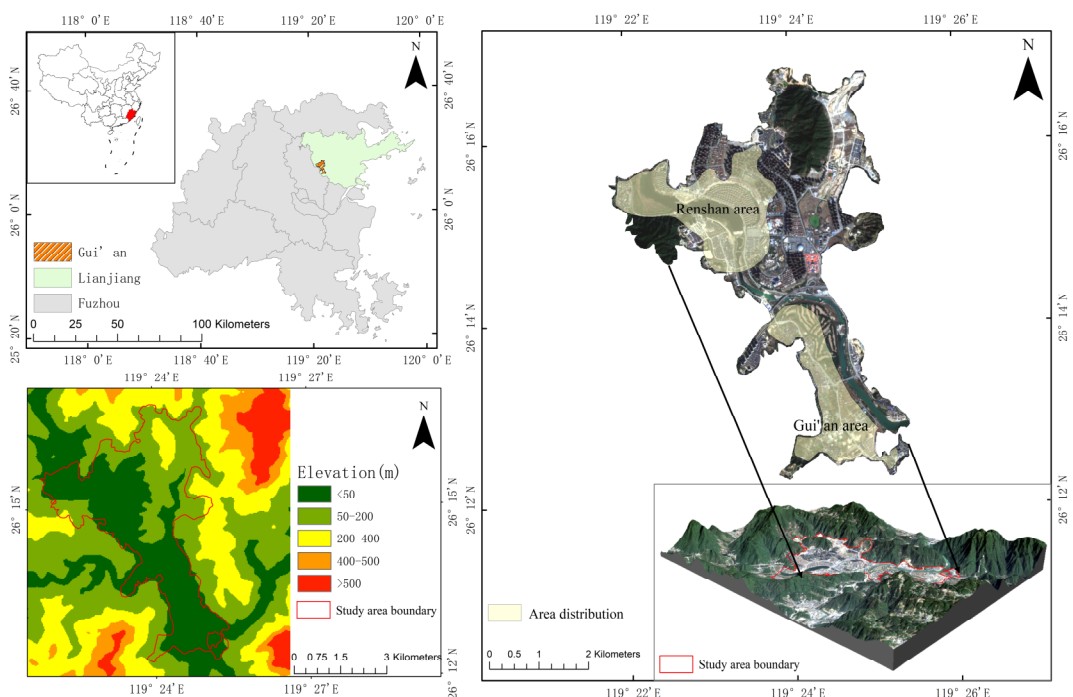

**Figure 1.** Location of the study area.

**Table 1.** Data source information.

| The Data Type | Image Time | Row Number | Spatial Resolution |
|---|---|---|---|
| Landsat 5 | 24 May 2010 | 119/042 | 30 m |
| Landsat 8 | 4 August 2013 | | |
| Sentinel-2A | 25 January 2016 | | 10 m |
| | 29 January 2019 | | |
| | 18 January 2021 | | |
| DEM | 2009 | | 30 m |

*2.2. Research Methods*

2.2.1. Optimization of Random Forest Classification

In order to find the most effective and optimal land classification method for complex topographic areas, reference [17] and Zhao Danping et al. [18] compared four machine learning methods including artificial neural network, decision tree, support vector machine and random forest, and found that the classification accuracy of random forest and decision tree was significantly higher than that of support vector machine and artificial neural network. The random forest method has the highest classification accuracy and good stability. It is proven that the random forest classification method can guarantee classification accuracy and obtain high classification efficiency at the same time, and is very effective for the extraction of land-use information in complex topographic areas. Data fusion can improve classification accuracy to a certain extent. Among many machine learning algorithms, the random forest classification method can show its advantages when the feature space is complex and the data present different statistical distribution [19]. Different researchers used random forest to study remote sensing data, and confirmed that this method effectively improved classification accuracy [20,21]. This paper classifies the

land in the region based on the random forest classification optimized using multi-source remote sensing data to monitor the land change dynamics in the study area more accurately (Figure 2).

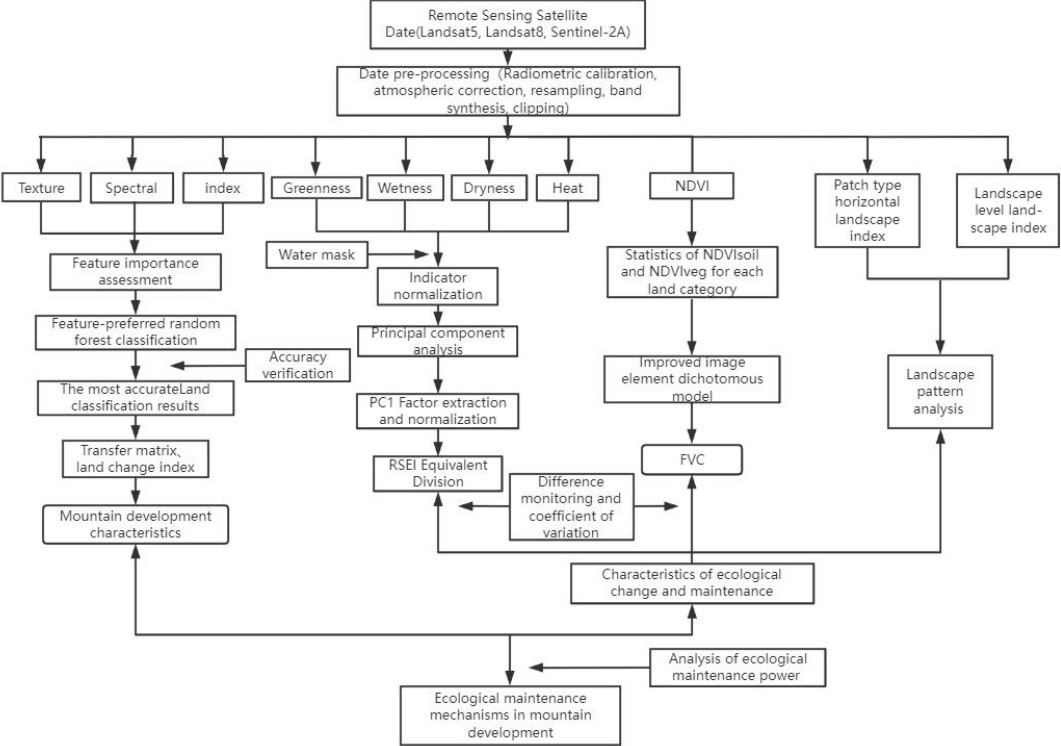

**Figure 2.** The flow chart of this study.

In view of the characteristics of data sources and research areas, and referring to previous research results, the rich spectral and spatial information of Landsat and Sentinel-2A data were fully utilized to optimize features through high-dimensional feature-space-dimensionality reduction to achieve high-precision land classification. Vegetation index and water index are more stable than single-band in reflecting land-cover types, which can improve the accuracy of image classification to a certain extent [22–24]. Additionally, the ratio operation of the band contained in the index can reduce the negative impact of the shadow caused by terrain. Texture features can reflect rich ground-object information and have been proven to improve the classification accuracy of images in medium- and high-resolution image classification [25,26]. In order to make full use of the sentinel-2A red-edge band of vegetation, five sentinel-2A red-edge indices were selected for land classification.

In this paper, 26 feature variables including spectral feature, texture feature and exponential feature were extracted from sentinel data to construct a feature space(Table 2). The importance ranking of each feature variable was obtained through repeated selection of feature variable classification, and the top 6, 10, 14, 20 and 26 feature variables were selected to implement the classification. A total of 14 feature variables, including spectral feature and texture feature, were selected for Landsat data(Table 3), and the top 5, 10 and 14 feature variables of importance were, respectively, selected for classification and optimization [27–29]. According to the national land classification and local experience, five land-use categories were determined; these are forest land (shrub, forest land, etc.), farm land (dry land, paddy field, irrigated land, etc.), built-up land (residential land, road, etc.), bare land, and water area (lake, pool, river, etc.). The selection of samples was based on field investigation results and Google Earth historical images, and the classification accuracy was improved by checking the classification results and modifying the samples.

**Table 2.** Sentinel data feature-type selection.

| Feature Type | Feature Variable | Abbreviation | Description |
|---|---|---|---|
| Spectral | Band | Band | b2, b3, b4, b5, b6, b7, b8, b8a, b11, b12 |
| Vegetation index | Normalized Difference Vegetation Index | NDVI | $(b8 - b4)/(b8 + b4)$ |
| | Ratio Vegetation Index | RVI | $b8/b4$ |
| | Difference Vegetation Index | DVI | $b8 - b4$ |
| Water index | Modified Normalized Difference Water Index | MNDWI | $(b3 - b11)/(b3 + b11)$ |
| Red edge index | Normalized Difference Vegetation Index1 | NDVIre1 | $(b8a - b5)/(b8a + b5)$ |
| | Normalized Difference Vegetation Index2 | NDVIre2 | $(b8a - b6)/(b8a + b6)$ |
| | Normalized Difference Vegetation Index3 | NDVIre3 | $(b8a - b7)/(b8a + b7)$ |
| | Normalized Difference 1 | NDre1 | $(b6 - b5)/(b6 + b5)$ |
| | Normalized Difference2 | NDre2 | $(b7 - b5)/(b7 + b5)$ |
| Texture | Mean | Mean | - |
| | Variance | Variance | - |
| | Contrast | Contrast | - |
| | Entropy | Entropy | - |
| | Correlation | Correlation | - |
| | Homogeneity | Homogeneity | - |
| | Dissimilarity | Dissimilarity | - |

**Table 3.** Landsat date feature type selection.

| Feature Type | Feature Variable | Abbreviation | Description |
|---|---|---|---|
| Vegetation index | Normalized Difference Vegetation Index | NDVI | $(NIR - R)/(NIR + R)$ |
| | Ratio Vegetation Index | RVI | $NIR/R$ |
| | Difference Vegetation Index | DVI | $NIR - R$ |
| Water index | Modified Normalized Difference Water Index | MNDWI | $(Green - MIR)/(Green + MIR)$ |
| Texture | Mean | Mean | - |
| | Variance | Variance | - |
| | Contrast | Contrast | - |
| | Entropy | Entropy | - |
| | Correlation | Correlation | - |
| | Homogeneity | Homogeneity | - |
| | Dissimilarity | Dissimilarity | - |
| Tassel cap change | Brightness | Brightness | - |
| | Greenness | Greenness | - |
| | Wetness | Wetness | - |

By adding each feature variable separately and repeating the operation of random forest classification, the random forest classification results, after adding different feature variables, were obtained. The accuracy of different classification results was verified using the error matrix, and the statistics of different kappa coefficients and classification accuracy were performed, as shown in Figure 3. Selecting the features that have the greatest influence on the results to build the model can not only reduce data redundancy, but also improve the accuracy of the model. Features with greater influence are more important. The purpose of determining the importance of features was simply to see how much contribution each feature made to each tree in the random forest. The average value was taken, and finally, the

contribution amount of each feature was compared. As can be seen in Figure 3, for sentinel data, the significance of modified normalized water index (*MNDWI*) features is the greatest, and the homogeneity of texture features is the lowest. For Landsat data, contrast was the most important feature among texture features, while mean was the least important.

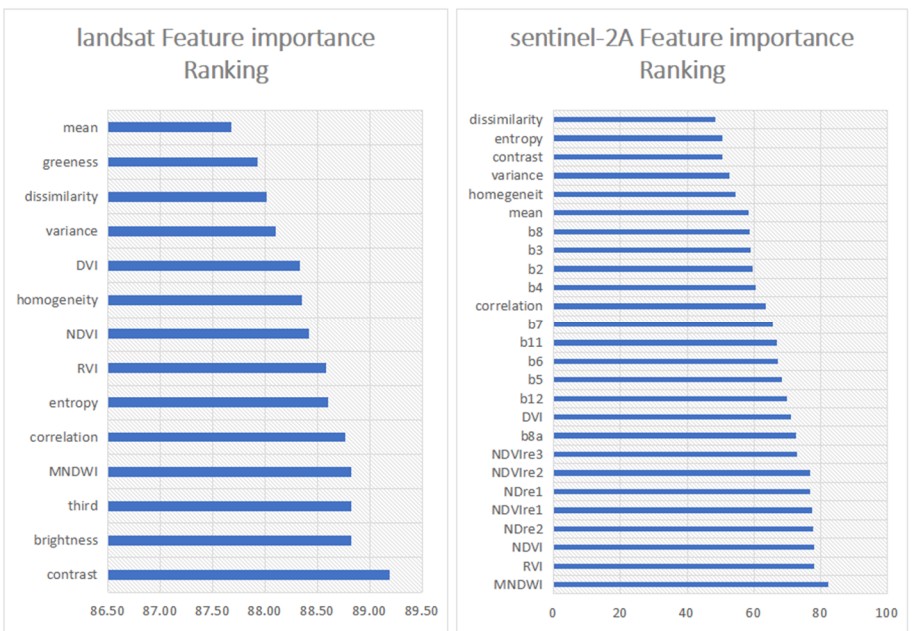

**Figure 3.** Feature-importance ranking.

The involvement of all features in a classification will lead to a redundancy of feature variables, a reduction in the image classification accuracy, and an increase in the operation cost; therefore, it is also necessary to select the optimal feature space for image classification. In this paper, we further classified the classified features by combining them according to their importance rankings, and then improved the classification speed and accuracy. For the Sentinel data, the top 6, 10, 14, 20 and 26 feature variables of importance were selected for classification, and for the Landsat data, the top 5, 10 and 14 feature variables of importance were selected for classification. The feature set with the maximum classification accuracy was chosen as the feature space for this final optimization [30]. From the results (Table 4), it can be seen that the classification accuracy increased first and then decreased as the feature variables increased. Among them, the Sentinel data scheme II (the top 10 feature variables selected for importance) was better classified with a kappa coefficient of 90.16%, while the Landsat data (the top five feature variables selected for importance) were better classified with a kappa coefficient of 90.11%. After the optimization of random forest classification by selecting feature variables, not only was the data redundancy reduced, but the classification accuracy was also greatly increased and better classification results were achieved.

**Table 4.** Comparison of accuracy of different solutions for Sentinel data.

| Image Data Type | Scheme | Number of Features | Kappa% |
|---|---|---|---|
| Sentinel-2A | 1 | 6 | 89.53 |
| | 2 | 10 | 90.16 |
| | 3 | 14 | 89.61 |
| | 4 | 20 | 89.56 |
| | 5 | 26 | 89.31 |
| Landsat | 1 | 5 | 90.11 |
| | 2 | 10 | 88.85 |
| | 3 | 14 | 89.35 |

### 2.2.2. Land-Use Change Index

The temporal land-use change was evaluated by calculating a single land-use dynamic attitude (*ARC*) with the change intensity index (*CII*) [31].

(1) *ARC* is a single land-use dynamic attitude. It is used to quantify the rate of change of the area of a specific land-use type in a certain period. Using ARC, the growth rate of land-use types from the initial year is normalized to facilitate the comparison of cyclical land-use changes. The calculation formula is:

$$ARC = \left[ \frac{A2 - A1}{A1} \right] \times \frac{1}{N} \times 100\%$$

In the equation, $A1$ and $A2$ are the area of a certain class at the beginning ($T1$) and the end ($T2$) of a certain period, respectively, and the number of years between them is $N$.

(2) The *CII* index takes into account the total area of the study area and calculates the ratio of the degree of change of each land-use type to demonstrate its overall rate of change. Higher positive *CII* values indicate rapid expansion, while negative values are considered as contraction, with higher absolute values indicating greater intensity of change. The calculation formula is:

$$CII = \left[ \frac{A2 - A1}{N} \right] \times \frac{1}{TA} \times 100\%$$

In the equation, $TA$ is the total area of the study area. Sun et al. [32] classified *CII* into very low ($|CII| < 0.1$), low ($0.1 \leq |CII| < 0.2$), medium ($0.2 \leq |CII| < 0.4$), fast ($0.4 \leq |CII| < 0.7$), and very fast ($|CII| \geq 0.7$) variations.

### 2.2.3. Analysis of the Transfer Matrix

The area of land in different periods was analyzed by counting the area of different land classes in each year, and the area transfer resulting from land-use changes was expressed in the form of a matrix. By analyzing the obtained transfer matrix, the mutual transformation between two temporally different land classes can be obtained, so that the changes in land classes and the characteristics of utilization can be specifically analyzed, and the expressions are as follows:

$$Sij = \begin{Bmatrix} S11 & S12 & \cdots & S1j \\ S21 & S22 & \cdots & S2j \\ \cdots & \cdots & \cdots & \cdots \\ Si1 & Si2 & \cdots & Sij \end{Bmatrix}$$

In the equation, *Sij* indicates the number of conversions of the *i*th land-use type to the *j*th land-use type in the study area.

### 2.2.4. Landscape Pattern Index

A landscape pattern is spatially represented as a mosaic of patches of different land-use types, reflecting the effect of the land's ecological process [33]. A landscape pattern is an effective means of revealing regional ecological status and spatial variation. Exploring the characteristics and evolution of land-use landscape patterns is an important way to profoundly understand the relationship between human activities and the natural environment, and an important breakthrough in achieving sustainable development [34,35]. Many scholars have used RS and GIS to study the landscape pattern of mountain areas, hills and other mountain areas, which has promoted research progress in related fields and methods. Svoray et al. [36] evaluated the most suitable landscape for development in different locations of Israeli cities. Buyantuyev et al. [37] studied the landscape change characteristics of Phoenix city and concluded that landscape characteristics have an important impact on vegetation and the social economy. Five patch types, namely patch number (NP), landscape proportion (PLAND), maximum patch index (LPI), landscape shape index (LSI), diversity index (SHDI), evenness index (SHEI), maximum patch index (LPI), and

patch density (PD) were selected to analyze the change in landscape pattern in the study area from different levels.

### 2.2.5. Remote Sensing Ecological Index Model

The Remote sensing ecological index (*RSEI*) is an index proposed by Xu Hanqiu based on remote sensing information technology, and can rapidly monitor and evaluate regional ecological quality [38,39]. Professor Xu Hanqiu found the comparability between the two by comparing the calculated results of the ecological index (*EI*) in The Technical Plan for Ecological Environment Status Assessment of the Ministry of Environmental Protection. However, the difference is that *RSEI* can not only serve as a quantitative index, but also carry out visualization, spatio-temporal analysis, and modeling and prediction of regional changes in the ecological environment. These can make up for the lack of *EI* index.

In addition, Yang Huiting [40] used the remote sensing ecological index (*RSEI*) and vegetation coverage (*FVC*) to evaluate vegetation coverage and ecological quality changes in the Wuyi Mountain Nature Reserve. Li Tingting [41] used the remote sensing ecological index (*RSEI*) to study the mountain ecosystem of the Helan Mountain. The results showed that the *RSEI*—obtained by coupling *WET*, *NDVI*, *SI* and *LST*—can reflect ecological quality comprehensively, and has a considerable effect on the monitoring of ecological quality in mountain areas.

In terms of remote sensing technology, it can obtain the information of 4 indicators from remote sensing images; these are vegetation index, bare soil index, humidity component and surface temperature, which can represent greenness, dryness, humidity and heat, respectively. In this way, the proposed remote sensing ecological index (*RSEI*) can be expressed as a function of these 4 indexes (Figure 2), namely:

$$RSEI = f(NDVI, Wet, LST, NDSI)$$

(1) Greenness index (*NDVI*)

The normalized vegetation index *NDVI* is the most widely used vegetation index, and is closely related to plant biomass, leaf area index and vegetation cover; therefore, *NDVI* was chosen to represent the greenness index, and the formula is:

$$NDVI = (NIR - R)/(NIR + R)$$

(2) Humidity index (*Wet*)

The brightness, greenness and humidity components obtained using remote sensing tasseled cap transformation have been widely used in ecological environment monitoring, among which the humidity component reflects the humidity of water bodies, soil and vegetation; there are closely related to ecology and, therefore, the humidity index was represented by the humidity component. The specific extraction equations of moisture components based on Landsat5 TM remote sensing data, Landsat8 OLI remote sensing data and Sentinel-2A remote sensing data are the following [1–3] equations [42,43]:

Landsat5:

$$Wet = (0.0315 \times float(b1) + 0.0201 \times float(b2) + 0.3102 \times float(b3) \\ + 0.1594 \times float(b4) - 0.6806 \times float(b5) \\ - 0.6109 \times float(b7))/10000 \tag{1}$$

Landsat8:

$$Wet = (0.1511 \times float(b2) + 0.1973 \times float(b3) + 0.3283 \times float(b4) \\ + 0.3407 \times float(b5) - 0.7117 \times float(b6) \\ - 0.4559 \times (b7))/10000 \tag{2}$$

Sentinel-2A:

$$Wet = 0.1509 \times b2 + 0.1973 \times b3 + 0.3279 \times b4 + 0.3406 \times b8 \\ - 0.7112 \times b11 - 0.4572 \times b12 \tag{3}$$

(3)  Dryness index

The selected index was the bare soil index (*SI*), but in the regional environment, there is a considerable built-up part, which also causes the "dryness" of the surface; thus, the dryness index can be synthesized from both, i.e., from the bare soil index (*SI*) and the building index (*IBI*) [44].

$$IBI = \frac{\frac{2\rho_{swir1}}{\rho_{swir1}+\rho_{nir}} - \left[\frac{\rho_{nir}}{\rho_{nir}+\rho_{red}} + \frac{\rho_{green}}{\rho_{green}+\rho_{swir1}}\right]}{\frac{2\rho_{swir1}}{\rho_{swir1}+\rho_{nir}} + \left[\frac{\rho_{nir}}{\rho_{nir}+\rho_{red}} + \frac{\rho_{green}}{\rho_{green}+\rho_{swir1}}\right]}$$

$$SI = \frac{[(\rho_{swir1} + \rho_{red}) - (\rho_{nir} + \rho_{blue})]}{[(\rho_{swir1} + \rho_{red}) + (\rho_{nir} + \rho_{blue})]}$$

$$NDSI = \frac{IBI + SI}{2}$$

(4)  Heat index (*LST*)

The ground surface temperature (*LST*) is one of the important physical indicators reflecting the natural ecological condition of Earth's surface, and is of great significance to the ecological environment; thus, the heat index was represented by the ground surface temperature, which was realized using the single window algorithm with the atmospheric correction method [45–47]. The calculation formula is as follows:

$$LST = \{a(1 - C - D) + [b(1 - C - D) + C + D]Tb - DTa\}/C$$

$$C = \tau\varepsilon$$

$$D = (1 - \tau)[1 + (1 - \varepsilon)\tau]$$

where *LST* is the inverse surface temperature; *Tb* is the brightness temperature (*K*) obtained by the sensor; *Ta* is the average atmospheric action temperature (*K*); *a* and *b* are regression coefficients; *C* and *D* are intermediate variables; $\varepsilon$ is the surface-specific emissivity; and $\tau$ is the atmospheric transmittance.

Due to the presence of a large amount of water in the study area, which can affect the moisture components and, thus, the loading values of the moisture components of the principal component analysis, a masking process was performed for the water bodies in the study area. To ensure the uniformity of the magnitudes, each indicator was regularized before conducting the principal component analysis. Finally, the four indicators were synthesized using principal component analysis to obtain the remote sensing ecological environment index. The greatest advantage of using principal component transformation for index integration is that the weights of the integrated indicators are not artificially determined, but are automatically and objectively determined according to the nature of the data themselves and the contribution of each indicator to each principal component; thus, it avoids the bias of the results caused by the weight setting, which varies from person to person and from method to method.

2.2.6. Fractional Vegetation Cover

Vegetation is an important link between the soil, moisture, organisms and atmosphere; it is an active member of the Earth system, which is very sensitive to the response of environmental changes [48]. The normalized vegetation index is more widely used in evaluating the environmental quality of terrestrial ecosystems and regulating ecological processes, and the evaluation results are more realistic and reliable [49]. Vegetation cover

is commonly used in vegetation change, ecological and environmental research, soil and water conservation, climate research, etc. It is important to reveal the characteristics of ecosystem change. Therefore, in this paper, the vegetation cover was calculated using an improved meta-dichotomous model [50] to analyze the vegetation changes in the study area at each stage. The formula is:

$$FVC = \frac{NDVI - NDVIsoil}{NDVIveg - NDVIsoil} \tag{4}$$

In the equation, *NDVI* is the image vegetation cover, *NDVIsoil* and *NDVIveg* are parameter files generated according to the minimum and maximum value of *NDVI* effective values corresponding to different land classes.

### 2.2.7. Coefficient of Variation

In this paper, the coefficient of variation was used to analyze the fluctuations of *RSEI* and *FVC* in 2010–2021. The coefficient of variation is often used to indicate the degree of fluctuation of time-series data; it can also indicate the degree of vulnerability of regional ecosystems to some extent [51], and it can quantify the degree of dispersion of a set of data for describing stability characteristics [52,53]. By calculating and visualizing the variation coefficient of the ecological remote sensing index (*RSEI*) and vegetation coverage (*FVC*) [54–56], we can intuitively observe the fluctuation degree and its distribution in a certain period of time, and reflect the maintenance status of ecological quality on the side. The calculation formula is:

$$CV = \frac{1}{\overline{x}} \sqrt{\frac{1}{n-1} \sum_{i=1}^{n} (xi - \overline{x})^2} \tag{5}$$

*CV* is the coefficient of variation, i is the time series, and *xi* is the image element value in year *I*; it is the mean value of the image element value in all years in the study period, and the interannual *RSEI (FVC)* of the image element is used as the input value. The larger value of the coefficient of variation indicates that the ecological environment is more fragile due to drastic changes, while the smaller value of the coefficient of variation indicates that the fluctuation is smaller, the ecological environment is better maintained, and the ecological environment is more resistant to disturbance.

### 2.2.8. Difference Calculation

Difference calculation is the subtraction of images with different time phases in the same region, using the difference between images to measure the magnitude of the change [57]. By calculating the difference between the remote sensing ecological index and vegetation coverage [58], the change in the ecological index in each stage can be accurately obtained. The formula is:

$$Dij = A_{ij}^{t1} - A_{ij}^{t2} \tag{6}$$

*Dij* is the difference in *RSEI (FVC)* in row *i* and column *j*, *Aij* is the image element value of *RSEI (FVC)* in row *i* and column *j*, and *t*1 and *t*2 are the time phases.

## 3. Results

### 3.1. Characteristics of Mountain Development Pattern

It can be seen in Figures 4 and 5 and Table 5 that Gui 'an has obvious changes in 2010–2016 and relatively stable changes in 2016–2021. According to the land classification results in Figure 4, the development and construction in Gui 'an from 2010 to 2021 are mainly concentrated in Renshan area in the north. The period from 2010 to 2013, characterised by mountain land reclamation and a huge amount of engineering, is the initial stage of Gui 'an development, and is also a crucial step. As can be seen from Table 5, land changes during this period are concentrated on the decrease in forest land and farm land, and the

increase in built-up and bare land. During this period, forest land and farm land decrease by 6.34 km², while built-up land increases by 4.47 km² and bare land increases by 2.67 km². It can be seen from the transfer matrix of this period that forest land and farm land are converted into built-up and bare land for construction. From 2013 to 2016, the forest land basically remains unchanged, and the changes are mainly concentrated on the decrease in bare land and the increase in built-up land. It can be seen that this period is the main construction stage; the forest land is basically unchanged, and the infrastructure, social system and leisure and entertainment industry are constantly improving and building. During this period, the bare land decreases by about 1.88 km², which is the main decrease. From 2016 to 2019, the land change is concentrated on the decrease in forest land and the increase in bare land. The decrease in forest land is about 0.91 km², and the increase in bare land is about 1.03 km², basically a small change. This period is Gui 'an's second land expansion stage. From 2019 to 2021, land expansion gradually tends to saturation with little change, and land change is concentrated in the increase in built-up land and the decrease in bare land; this is the second construction period dominated by the increase in built-up land. According to the results, the development stage of Gui 'an can be roughly divided into two development and construction stages: 2010–2016 is the first stage of development and construction, and 2016–2021 is the second stage of development and construction.

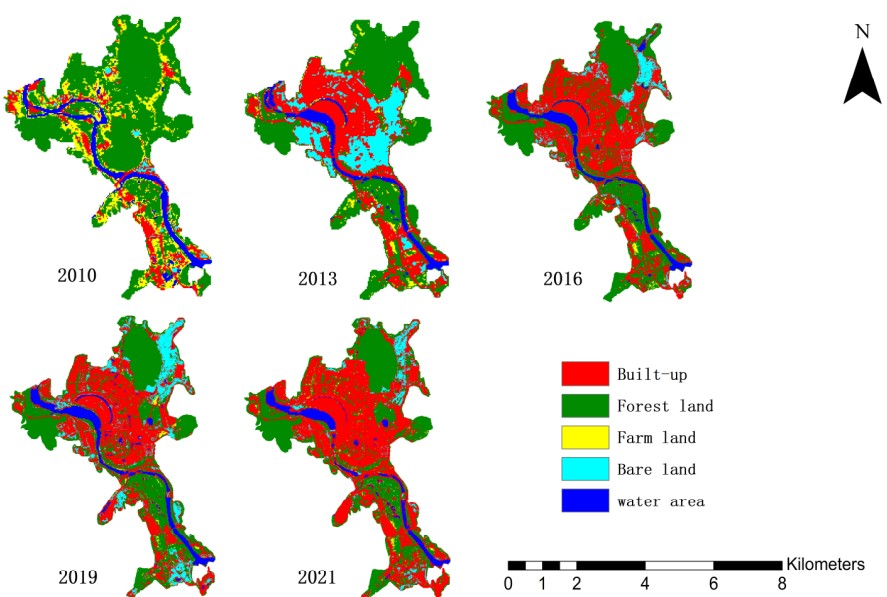

**Figure 4.** Land classification results for 2010–2021.

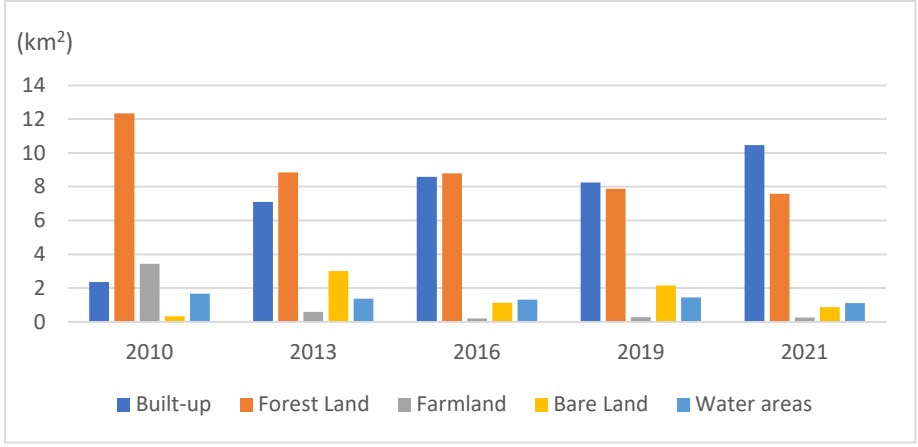

**Figure 5.** Folded map of land area change.

**Table 5.** Land-use area transfer matrix 2010–2021 (percent).

| Year (Latter\Early) | | Built-Up | Water Areas | Bare Land | Farmland | Forest Land |
|---|---|---|---|---|---|---|
| 2010–2013 | Built-up | 71.89 | 25.20 | 74.41 | 43.47 | 25.94 |
| | Water areas | 4.10 | 53.75 | 2.61 | 3.35 | 1.93 |
| | Bare land | 6.40 | 1.67 | 13.58 | 17.71 | 17.61 |
| | Farmland | 2.57 | 2.16 | 1.57 | 5.89 | 2.31 |
| | Forest land | 15.05 | 17.22 | 7.83 | 29.58 | 52.23 |
| 2013–2016 | Built-up | 65.69 | 4.96 | 76.53 | 24.33 | 14.65 |
| | Water areas | 1.45 | 76.58 | 0.72 | 1.94 | 1.35 |
| | Bare land | 4.35 | 0 | 7.46 | 8.51 | 6.63 |
| | Farmland | 0.61 | 0.13 | 0.36 | 9.41 | 0.90 |
| | Forest land | 26.39 | 16.12 | 14.58 | 50.60 | 66.55 |
| 2016–2019 | Built-up | 72.74 | 5.98 | 36.34 | 10.00 | 17.01 |
| | Water areas | 1.29 | 84.98 | 1.01 | 1.74 | 2.37 |
| | Bare land | 10.52 | 4.52 | 50.04 | 8.55 | 7.02 |
| | Farmland | 0.91 | 0.66 | 3.16 | 14.73 | 1.46 |
| | Forest land | 14.54 | 3.86 | 9.45 | 64.98 | 72.14 |
| 2019–2021 | Built-up | 89.48 | 29.34 | 50.64 | 37.03 | 17.74 |
| | Water areas | 0.61 | 68.72 | 0.19 | 3.42 | 0.62 |
| | Bare land | 1.92 | 1.06 | 27.58 | 4.13 | 1.22 |
| | Farmland | 0.35 | 0.01 | 0.30 | 12.56 | 2.37 |
| | Forest land | 7.64 | 0.86 | 21.29 | 42.87 | 78.05 |

According to the analysis of the land change index, it can be seen from Figures 6 and 7 that the intensity and speed of change of each land type are great from 2010 to 2013, at the maximum value in the study period, realizing the purpose of rapid development and construction. The change rate of bare land reached its maximum. The change intensity and speed of forest land are −5.7% and −9.5%, respectively. The change intensity and speed of farm land are −4.6% and −27.5%, respectively. Because the development of forest land is more difficult than that of farm land, the change rate is smaller than that of farm land. From 2013 to 2016, bare land rapidly decreases in a large area, with a change intensity of −3.1% and a change rate of −20.8%. Built-up land increased, with a change intensity of 2.4% and a change speed of 7%. Since the first stage of development (2010–2013), the intensity and rate of change of most land types decrease. The period of 2016–2019 is the development period of the second stage, during which the bare land continues to grow, and the change intensity and rate decrease significantly compared with that of the first stage, at 1.7% and 30.4%, respectively. The change intensity and speed of forest land are −1.5% and −3.5%, respectively, while the change in farm land is small. In the second construction period (2019–2021), forest land and farm land remain stable, while bare land decreases and built-up land increases. The change intensity and speed of built-up land are 5.5% and 13.4%, respectively, slightly higher than that of the first construction period. During the study period, the water area always fluctuates gently, with little change.

Therefore, through comprehensive analysis, Gui'an adopts the "two-stage" mode to rapidly develop and recover, so as to reduce the impact on ecology brought about by the development process, and the boundary of the city has not changed from 2013 to 2021. Land outside the boundary is non-built-up, such as agricultural land, water, and forest land, and needs to be strictly protected. The delineation of the urban growth boundary limits the disorderly spread of urban space; protects the regional ecological space and maintains a certain regional balance between urban built-up and non-built-up land; realizes a high-density and more compact development pattern of the city; and efficiently utilizes the land within the existing urban area and the fringe areas. Additionally, Gui'an uses urban space intensively, especially the most rapid development of the Renshan area. As shown in Figure 8, From 2010 to 2021, the Renshan area—as the area with the most drastic changes in the class within the time period—contains 15 major supporting facilities

including Gui 'an Happy World, the Flower Expansion Park, the People's Promenade, five-star hotels, shopping centers, traditional cultural streets, and residential communities, to solve the integration of education, medical care, living and employment. Through the strict planning of land use, overall optimization of urban space is achieved, thus realizing the efficient use of land.

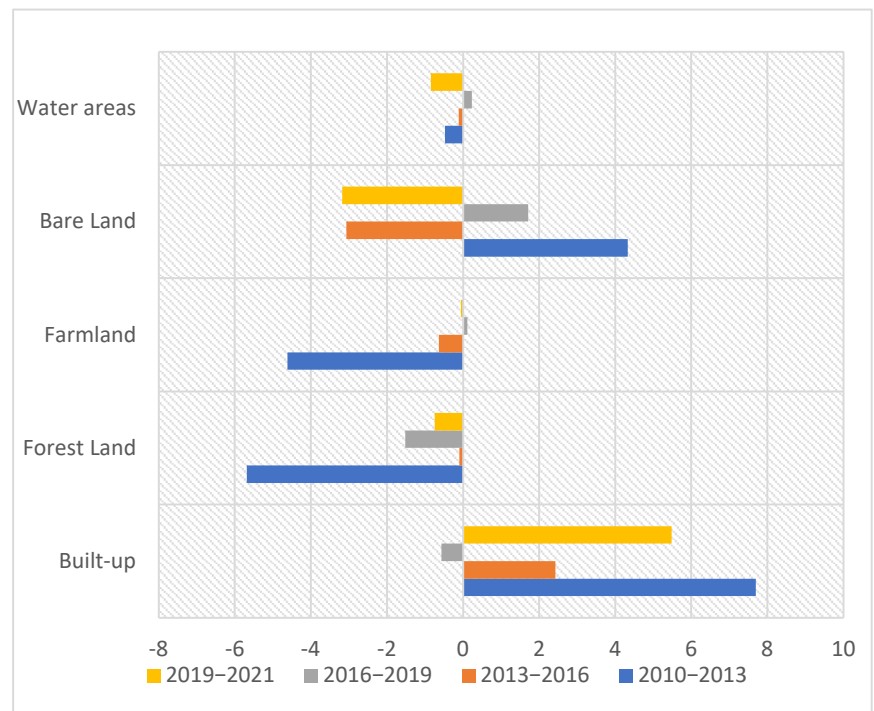

**Figure 6.** Change intensity (CII).

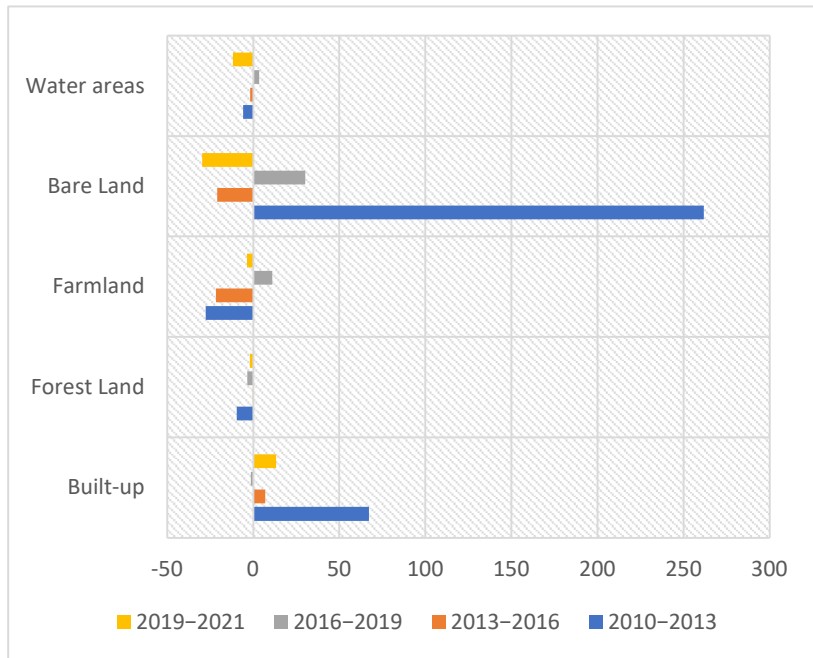

**Figure 7.** Rate of change (ARC).

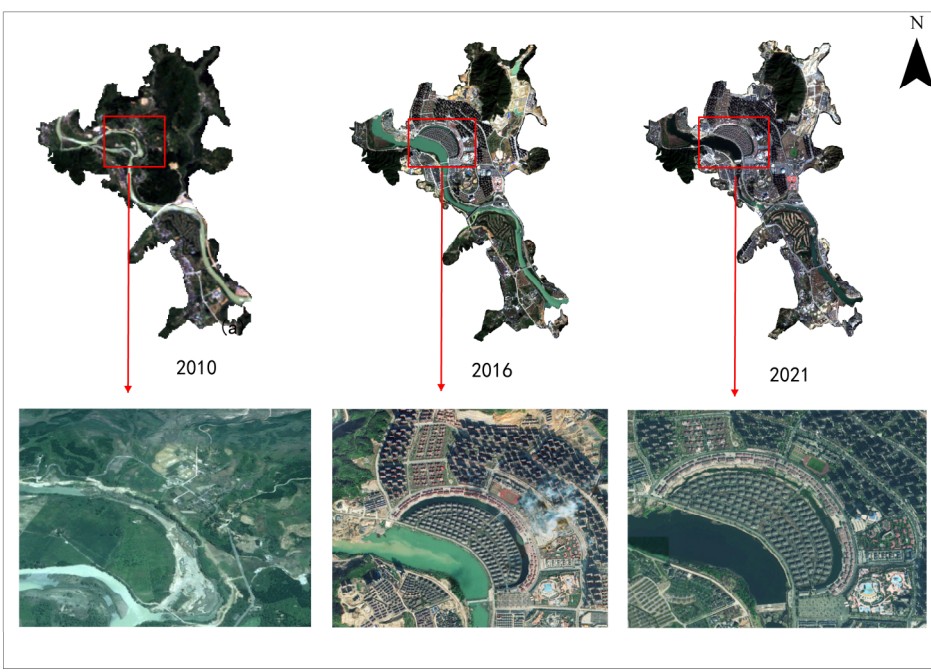

**Figure 8.** Specific urban construction changes in Gui 'an during the two-stages.

### 3.2. Analysis of Spatial and Temporal Changes in Ecological Maintenance

(1) To achieve specific analysis of the *RSEI*, levels were divided into: 0–0.2 for the lower level, 0.2–04 for the low level, 0.4–0.6 for the medium level, 0.6–0.8 for the high level, and 0.8–1 for the higher level of the five levels, as shown in Figure 9 and Table 6. It can be seen that the area with the greatest fluctuation of ecological quality is mainly the Renshan area, except for the main bare land, which is of poor grade, and most of the built-up land is of medium or poor grade. By monitoring the difference in ecological index, it was found that the characteristics of ecological change are closely related to the development stage of the land. As can be seen in Figure 10 and Table 7, the average value of the ecological index in the period from 2010 to 2021 shows a general downward trend. Perticularly during the period of development and expansion from 2010 to 2013, the ecological area of excellent grade decreases greatly, accounting for 45.64% of the total area. Large areas of land are bare, and the ecological status declines, especially in the Renshan Area. The concentrated development of commercial, cultural and tourism projects has greatly reduced the area with excellent ecological quality, and has reduced the ecological quality; however, the reduction degree is always controlled below the drastic reduction degree, and the reduction degree is mainly moderate. In the late stage of the first stage, the ecological quality remains unchanged, accounting for 4.82%. The declining area decreases from 45.64% to 35.04%, and the ecological decline trend is controlled. In the second development stage from 2016 to 2021, the area of ecological decline remains unchanged. In the first half of the period, the area of ecological decline is significantly reduced, accounting for only 16.48%. There is even a phenomenon of ecological quality recovery, with the area increasing from 18.14% to 27.94%. In the latter half of the period, the ecosystem is stable as a whole. The second development and construction stage of Gui 'an basically completed the construction and development needed for economic development, and gradually paid attention to ecological construction and maintenance. During this period, the ecology was well maintained. During development, attention was paid to the construction and protection of urban ecological green space. In addition, due to urban construction, the ground was hardened, and soil erosion in mountain areas was reduced to a certain extent. The ecological grade gradually improved from poor to medium, and the areas of medium, good and excellent ecological quality grades were larger than those of poor and poorer grades.

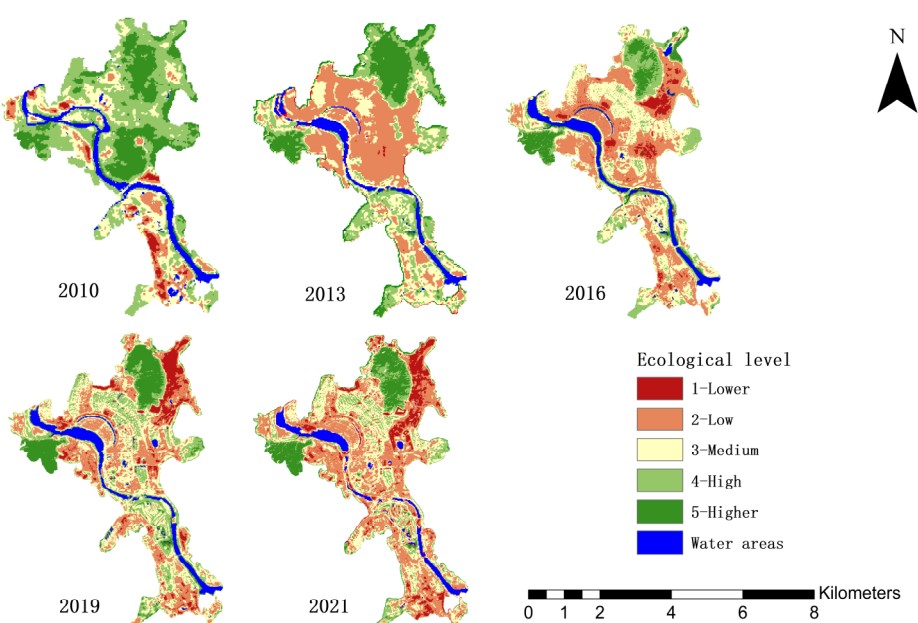

**Figure 9.** Ecological class distribution and change.

**Table 6.** RSEI level analysis statistics.

| RSEI Level | 2010 | | 2013 | | 2016 | | 2019 | | 2021 | |
|---|---|---|---|---|---|---|---|---|---|---|
| | Area km² | Scale% | Area km² | Scale% | Area km² | Scale% | Area km² | Scale% | Area km² | Scale% |
| Lower | 0.45 | 2.45 | 0.0675 | 0.36 | 0.7122 | 3.92 | 1.3607 | 7.34 | 1.5738 | 8.44 |
| Low | 1.8459 | 10.05 | 6.9192 | 36.66 | 6.5334 | 35.96 | 5.5576 | 29.97 | 6.903 | 37.03 |
| Medium | 4.0671 | 22.15 | 4.923 | 26.08 | 6.4388 | 35.44 | 5.8897 | 31.76 | 6.036 | 32.38 |
| High | 7.9434 | 43.25 | 4.2039 | 22.27 | 3.0807 | 16.96 | 3.6833 | 19.86 | 2.408 | 12.92 |
| Higher | 4.059 | 22.10 | 2.7603 | 14.62 | 1.4022 | 7.72 | 2.0511 | 11.06 | 1.723 | 9.24 |

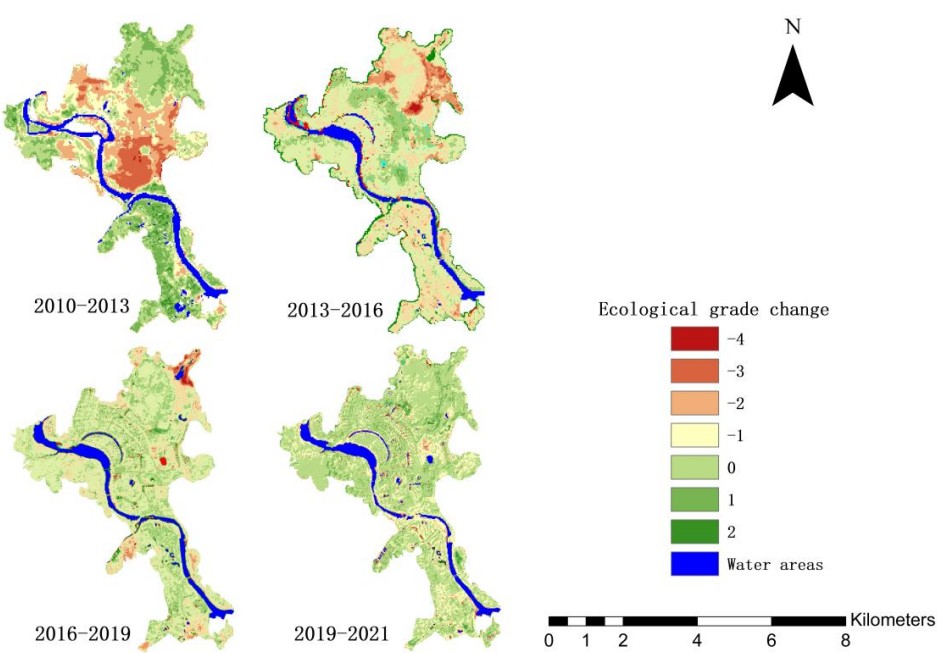

**Figure 10.** Ecological grade change map.

**Table 7.** Statistics on the change of RSEI from 2010 to 2013.

|  | Category | Level Difference | Level Area | Class Area | Scale% |
|---|---|---|---|---|---|
| **2010–2013** | Falling | −4 | 0.0279 | 8.2026 | 45.64 |
|  |  | −3 | 1.4778 |  |  |
|  |  | −2 | 3.1833 |  |  |
|  |  | −1 | 3.5136 |  |  |
|  | Unchanged | 0 | 6.084 | 6.084 | 33.85 |
|  | Rising | 1 | 3.4515 | 3.6864 | 20.51 |
|  |  | 2 | 0.2349 |  |  |
|  |  | 3 | — |  |  |
| **2013–2016** | Falling | −4 | 0.054 | 6.3216 | 35.04 |
|  |  | −3 | 0.2916 |  |  |
|  |  | −2 | 1.2357 |  |  |
|  |  | −1 | 4.7403 |  |  |
|  | Unchanged | 0 | 8.4456 | 8.4456 | 46.82 |
|  | Rising | 1 | 2.9619 | 3.2724 | 18.14 |
|  |  | 2 | 0.3015 |  |  |
|  |  | 3 | 0.009 |  |  |
| **2016–2019** | Falling | −4 | 0.0444 | 2.9669 | 16.48 |
|  |  | −3 | 0.1089 |  |  |
|  |  | −2 | 0.4288 |  |  |
|  |  | −1 | 2.3848 |  |  |
|  | Unchanged | 0 | 10.0074 | 10.0074 | 55.58 |
|  | Rising | 1 | 4.819 | 5.0312 | 27.94 |
|  |  | 2 | 0.2003 |  |  |
|  |  | 3 | 0.0119 |  |  |
| **2019–2021** | Falling | −4 | 0.0019 | 5.0522 | 27.56 |
|  |  | −3 | 0.0373 |  |  |
|  |  | −2 | 0.4631 |  |  |
|  |  | −1 | 4.5499 |  |  |
|  | Unchanged | 0 | 11.3518 | 11.3518 | 61.93 |
|  | Rising | 1 | 1.8435 | 1.9264 | 10.51 |
|  |  | 2 | 0.0829 |  |  |
|  |  | 3 | — |  |  |

As can be seen from the analysis, the "two-stage" development model reduces the impact of mountain development on ecological quality to a large extent, and combines urban construction and ecological environment maintenance in each stage of construction to avoid serious degradation of ecological quality. In the first stage of development, the urban growth boundary is defined according to the natural limit elements, and the two forest land areas in the northeast and west of the Renshan area, with an elevation of about 200–400 m, are relatively large and rich in vegetation cover, so they have not been developed and built as ecological protection areas. The natural landscape is maintained as a landscape skeleton in the overall contour of the mountainous city and an ecological limit to the development of urban spatial forms. Additionally, different functional areas are divided according to

the characteristics of resources and the fragility of the ecological environment in order to reduce the degree of ecological damage which, in absolute terms, contributes to ecological maintenance.

(2) The analysis of ecological fluctuations by calculating the coefficient of variation divides the rank into high fluctuating changes ($CV \geq 0.3$), relatively high fluctuating changes ($0.2 \leq CV \leq 0.3$), medium fluctuating changes ($0.15 \leq CV \leq 0.2$), relatively low fluctuating changes ($0.15 \leq CV \leq 0.1$), and low fluctuating changes ($CV \leq 0.1$). The lower the coefficient of variation, the lower the fluctuation of land change. It can be seen in Figure 11 that in 2010–2021, the two forest lands in the northeast and west direction of the Renshan Area are always the most ecologically protected areas that have not been developed; they have low fluctuating changes, and the ecology is well maintained. In the central part of the area, ecological fluctuation is also relatively low due to the urban greening construction. In the Gui 'an area, most of the forest lands remain below the relatively high fluctuation. However, in the construction areas of the Renshan area and Gui 'an area, due to the changes in forest land and farm land, the area of bare land increases, the ecological fluctuation is high, and the ecology is more fragile. In the Gui 'an area, half of the area remains in low and lower fluctuations, and the ecology is generally good.

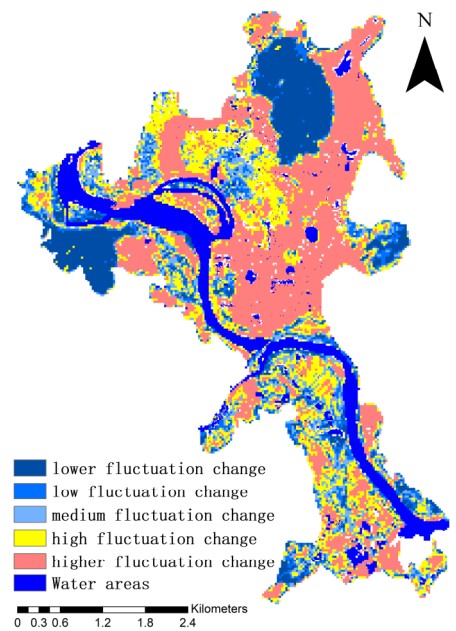

**Figure 11.** RSEI Fluctuations.

(3) From the correlation analysis of *RSEI, LST* and *FVC*, it can be seen in Figure 12 that there is a high correlation between *RSE*I, *LST* and *FVC*. Except for a few points, *RSEI* and *FVC* show a positive correlation, and the ecological index shows an increasing trend with the increase in vegetation cover. Additionally, the surface temperature showed a negative correlation with both the ecological index and vegetation cover, and the ecological index and vegetation cover showed a decreasing trend with the increase in temperature value. Therefore, vegetation cover plays an important role in the ecological maintenance and improvement of the region.

As shown in Figure 13, in the analysis of the change in vegetation cover from 2010 to 2013, the area of decreasing vegetation cover is larger and the area of increasing area is smaller. The area of decreasing vegetation cover is mainly located in the newly built Gui 'an New Area, which was in the construction stage in 2013, resulting in a large area of bare ground surface and a large area of vegetation damage. The period of 2013–2016 is the second stage of the construction period, and the vegetation cover is unchanged. From 2013 to 2016, the vegetation cover is mainly unchanged, and the area of rising vegetation cover increases greatly, while the area of declining vegetation cover is smaller, indicating that

the vegetation cover rebounded significantly. The area of rising vegetation cover is mainly located in Gui 'an New Area, which was completed in 2016, and its function is positioned as a hot-spring-themed integrated tourism resort, integrating hot-spring tourism, health and recuperation, and other multi-functional functions; this attaches importance to the construction of a green landscape, and the vegetation cover rebounded greatly compared with that in 2013. In 2016–2019, the *FVC* grade mostly unchanged, and the increasing area of vegetation coverage was slightly smaller than the decreasing area. This period is the second round of land development and expansion of Gui 'an, and also maintains the ecological rebound, increasing the construction of urban green space, so the vegetation is better-maintained. The perios 2019–2021, the second stage of construction period, remains mainly unchanged; the rising vegetation area undergoes a rebound phenomenon, the change gradually stabilizes, the fluctuation decreases, and the construction of Gui 'an basically ends.

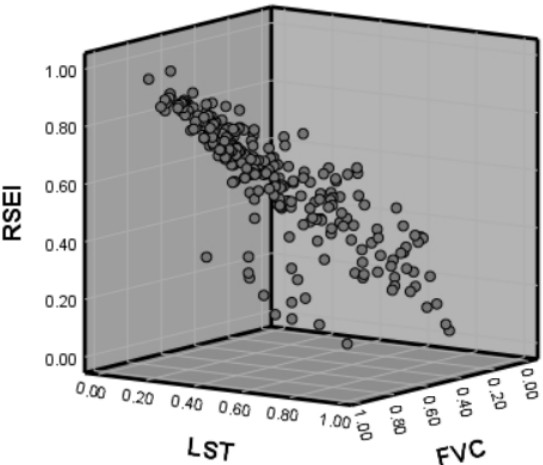

**Figure 12.** Three-dimensional spatial scatter plot.

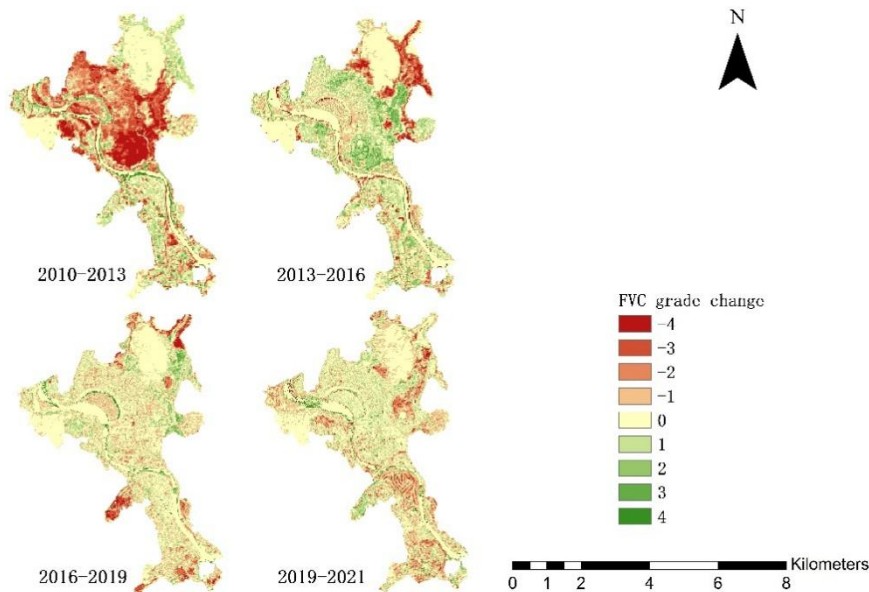

**Figure 13.** FVC grade change chart.

(4) The fluctuation status of vegetation was analyzed using the coefficient of variation of vegetation cover in the study area, and Yueyue Huang et al. classified the *CV* coefficient of vegetation cover into high fluctuation variation ($CV \geq 0.3$), relatively high fluctuation

variation ($0.2 \leq CV \leq 0.3$), medium fluctuation variation ($0.15 \leq CV \leq 0.2$), relatively low fluctuation variation ($0.15 \leq CV \leq 0.1$), and low volatile change ($CV \leq 0.1$).

It can be seen in Figure 14 that in this period of 2010–2021, there is a significant large area of vegetation reduction, which is due to the establishment of the less built-up Gui 'an resort area with the increase in bare land. However, the forest land in the western and northeastern part of the Renshan area always maintains better and low fluctuating changes. In the central part of the urban area and Gui 'an area, there are parts that maintain low fluctuation and lower fluctuation changes. The rest of the built-up land is mostly in a high fluctuation variation.

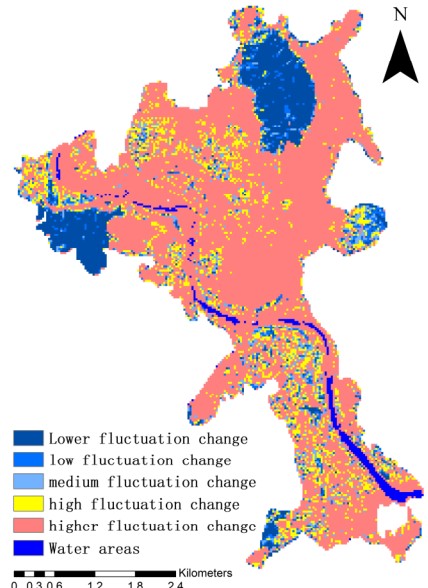

**Figure 14.** FVC Fluctuations.

(5) The change in land use in the process of urban development and construction results in a change in the landscape index. As shown in the horizontal landscape index of patch type in Table 8, the landscape proportion changes the most from 2010 to 2013. The landscape proportion of built-up and bare land increases significantly, while that of forest land and farm land decreases significantly. From 2013 to 2016, the proportion of forest land landscape increases to 43.88%, while the proportion of bare-land landscape decreases to 5.67%, and the increase rate of built-up land slows down. It can be seen that this period went from a development and expansion to a construction phase. From 2016 to 2019, Gui 'an started the second round of development and expansion. The proportion of bare land landscape increases, while the proportion of forest land decreases, but compared with the first round of development and expansion from 2010 to 2013, the change range is decreased. The period of 2019–2021 is the second round of construction in Gui 'an. The proportion of built-up land increases to 51.54%, the proportion of bare land decreases from 10.81% to 4.34%, and the change in forest land and farm land is relatively small. In general, with 2016 as the intermediate node, the change range from 2016 to 2021 is smaller than that from 2010 to 2016. As the Aojiang river, which flows through Gui 'an, is an important water-source-protection area in Lianjiang County, there is little change in water areas during the development period from 2010 to 2021.

Patch quantity (NP) and patch density (PD) show an increasing trend for each landscape type from 2010 to 2021. The change trend of forest land, built-up land and bare land is basically similar, with little change from 2010 to 2013; this is due to the rapid development mode of the whole large area in the early period of Gui 'an. The patch number (NP) and patch density (PD) increas significantly from 2013 to 2016, slows down from 2016 to 2019, and then decreases from 2019 to 2021. This indicates that in 2013–2016, the first stage of urban construction in Gui 'an, landscape patches of built-up land rapidly eroded

other landscape patches, presenting patchy characteristics and increasing the degree of fragmentation of forest land and built-up land. In 2016, Gui' an enters the second stage of development, and the range of change is relatively reduced. From 2019 to 2021, the landscape fragmentation degree of forest land, built-up land and bare land decreases due to the division and construction of the Gui' an functional zone, as well as the construction of ecological land and the intensive use of land. Due to the urban construction and transformation of a large amount of farm land, the NP and PD indices of farm land increase and the degree of fragmentation increases.

**Table 8.** Patche-type level landscape index.

| Year | Item | PLAND | NP | PD | LPI | LSI |
|------|------|-------|-----|-----|-----|-----|
| 2010 | Built-up | 11.66 | 169.00 | 8.39 | 3.47 | 18.08 |
| | Forest land | 61.24 | 124.00 | 6.16 | 39.52 | 14.23 |
| | Farm land | 17.09 | 387.00 | 19.21 | 1.47 | 28.61 |
| | Bare land | 1.71 | 95.00 | 4.72 | 0.36 | 10.33 |
| | Water area | 8.29 | 55.00 | 2.73 | 3.81 | 9.43 |
| 2013 | Built-up | 33.91 | 110.00 | 5.26 | 20.68 | 17.25 |
| | Forest land | 42.25 | 131.00 | 6.26 | 19.55 | 14.43 |
| | Farm land | 2.88 | 242.00 | 11.57 | 0.18 | 16.71 |
| | Bare land | 14.37 | 101.00 | 4.83 | 8.51 | 10.18 |
| | Water area | 6.59 | 34.00 | 1.62 | 2.43 | 7.82 |
| 2016 | Built-up | 42.87 | 645.00 | 32.24 | 39.26 | 40.87 |
| | Forest land | 43.88 | 1166.00 | 58.28 | 10.32 | 39.04 |
| | Farm land | 1.03 | 682.00 | 34.09 | 0.16 | 26.71 |
| | Bare land | 5.67 | 855.00 | 42.74 | 1.82 | 30.03 |
| | Water area | 6.55 | 169.00 | 8.45 | 2.68 | 11.00 |
| 2019 | Built-up | 41.20 | 927.00 | 46.34 | 29.94 | 45.22 |
| | Forest land | 39.35 | 1546.00 | 77.28 | 8.92 | 37.80 |
| | Farm land | 1.41 | 799.00 | 39.94 | 0.10 | 28.74 |
| | Bare land | 10.81 | 1201.00 | 60.03 | 4.35 | 37.10 |
| | Water area | 7.24 | 257.00 | 12.85 | 3.61 | 14.69 |
| 2021 | Built-up | 51.54 | 579.00 | 28.56 | 47.24 | 37.74 |
| | Forest land | 37.37 | 1228.00 | 60.58 | 8.57 | 36.36 |
| | Farm land | 1.28 | 956.00 | 47.16 | 0.09 | 33.19 |
| | Bare land | 4.34 | 640.00 | 31.57 | 1.28 | 28.81 |
| | Water area | 5.47 | 396.00 | 19.54 | 2.51 | 1.13 |

Prior to 2013, forest land has the largest maximum patch index (LPI), followed by built-up land. After 2013, built-up has the largest LPI index, followed by forest land. The LPI index of farm land, bare land and water areas is relatively small, and the range of change is not large. This indicates that with the development and construction of the city, built-up land gradually occupies a dominant position. However, as Gui 'an is a comprehensive ecological tourism city, a certain amount of forest land will be maintained, making the LPI index of forest land second only to that of built-up land.

Based on the analysis of the landscape shape index (LSI) of each landscape type, it can be seen that the LSI index of built-up land, forest land, water area and bare land basically remains stable from 2010 to 2013, while the LSI index of farm land decreases from 28.62 to 16.71. This is because Gui 'an defined the growth boundary and spatial planning scope of the city in the initial development stage; thus, the development is carried out in accordance with the planning scope in the development and expansion stage, avoiding ecological environment damage caused by excessive development and disorderly development to a large extent. From 2013 to 2016, the LSI index of all landscape types increases, and in 2016, the LSI index of built-up is the largest, followed by forest land, bare land, farm land and water area. The increase in the forest LSI index was beneficial to the improvement of biodiversity. From 2016 to 2021, the LSI index of all landscape types has a small change

range, indicating that land change became smaller during this period, and development and construction entered a relatively stable stage, mainly involving the regional development and construction of cities.

(6) As shown in Figure 15, the LPI, SHDI and SHEI indicators of Gui 'an are obviously divided into two similar change stages, with 2016 as the intermediate node. Additionally, the change range of the second stage is smaller than that of the first stage. During the two development and expansion periods of 2010–2013 and 2016–2019, SHEI increases, LPI decreases, the proportion of landscape types tends to be balanced, and the degree of distribution uniformity increase. SHDI increases, indicating that landscape heterogeneity increased, patch types tended to diversify, and landscape richness increased during development and expansion. In the two construction and conservation periods from 2013 to 2016 and 2019 to 2021, SHEI decreases while LPI increases, indicating that construction increased the differences among landscape types in the region and dominant patch types tended to be significant. SHDI decreases because planning and construction reduced the difference between regional landscape types and landscape heterogeneity.

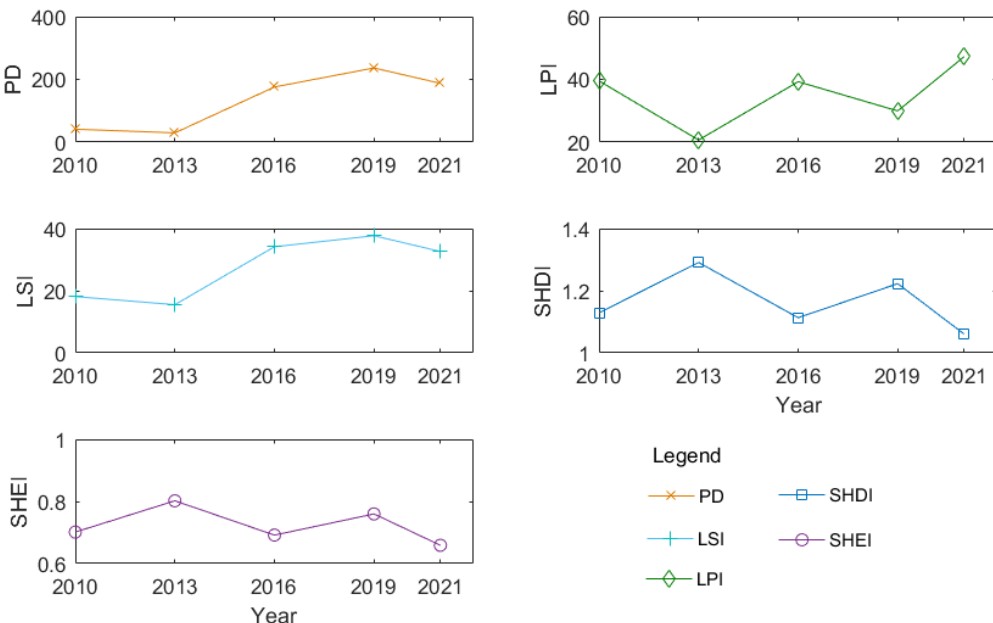

**Figure 15.** Landscape-level landscape index.

Landscape density (PD) and landscape shape index (LSI) have similar changes. Both show little change from 2010 to 2013, significantly increase from 2013 to 2016, and decreased from 2016. PD and LSI decline and tend to be stable in 2019. This indicates that during the development and expansion period of Gui 'an from 2010 to 2013, the rapid development of the whole wide range resulted in little change in landscape density and shape. During the construction period from 2013 to 2016, landscape density increases, landscape fragmentation increases, and landscape shape becomes more irregular and gradually more complicated. In the second stage from 2016 to 2021, the change decreases, and in the construction period from 2019 to 2021, the degree of landscape fragmentation decreases, and the landscape shape tends to be simplified under the influence of human activities.

## 4. Discussion

In the current management of ecological problems, ecological restoration is taken seriously. However, ecological restoration is a kind of post-compensation behavior that relies on the self-organization and self-regulation ability of the ecosystem and appropriate human guidance; this is necessary to curb the further degradation of the ecosystem under the condition that the ecological environment has been severely damaged and degraded by human activities. However, we must admit that preventing irreversible damage can

yield more benefits than trying to restore it later. Therefore, ecological maintenance is particularly important in today's environment, where the contradiction between economic development and ecological protection is increasingly fierce.

In this paper, the "two-stage" land development pattern of Gui 'an was summarized using the methods of random forest classification and change analysis. According to a comprehensive ecological-quality evaluation model based on the remote sensing ecological index, vegetation coverage, and the landscape index, the ecological quality change of the study area is closely related to the land development pattern. Table 9 summarizes the characteristics of land use change and ecological maintenance in Gui 'an in various periods. From the initial stage of development and construction to the stable development of a region, the effect of ecological maintenance largely depends on the choice of development mode in the development process. During the development process, various artificial guided measures should be taken to reduce the impact and damage of urban construction on the ecological environment. There is a greater overall benefit in preventing irreversible effects than in trying to recover afterwards. The "two-stage" land development mode of Gui 'an is an important reason for its ecological quality to be maintained at a certain level in the process of development and construction. Through summary and analysis, several factors affecting ecological sustainability can be concluded:

**Table 9.** Summary of land development change and ecological maintenance.

| Stage | Period | Time Range | Characteristics of Land Change | Ecological Maintenance |
|---|---|---|---|---|
| The first stage | Development and expansion period | 2010–2013 | Forest land and farm land decrease, while built-up and bare land increase (with great changes); the expansion strength is large and the change speed is fast | *RSEI* excellent grade decreases significantly, and the decline degree is mainly medium; *FVC* is mainly decreased; the heterogeneity and uniformity of landscape increase, but the degree of fragmentation and shape index do not change much. The two forest lands in the northeast and west of Renshan area have always been the most undeveloped ecological protection areas. The ecological quality fluctuates greatly in the middle of Renshan area |
| | Construction and conservation period | 2013–2016 | The forest land is basically unchanged, the bare land decreases and the built-up land increases; rapid construction of built-upland ; the intensity and rate of change are large | *RSEI* remains unchanged and the downward trend is controlled; *FVC* remains unchanged and rises; the uniformity and diversity of landscape decreases while the degree of fragmentation and shape index increase |
| The second stage | Development and expansion period | 2016–2019 | Mainly the forest land and bare land increase (change small compared with 2010–2013); the intensity and rate of change decrease | *RSEI* is mainly unchanged with the phenomenon of ecological quality recovery; *FVC* is mainly constant; the diversity and uniformity of landscape are increased, and the degree of fragmentation and complexity of landscape shape are controlled |
| | Construction and conservation period | 2019–2021 | Mainly an increase in built-up land and the decrease in bare land (small change); the intensity and rate of change decrease | *RSEI* remains unchanged and tends to be stable as a whole; *FVC* is mainly constant and tends to be stable; landscape diversity and uniformity decreases, fragmentation degree and shape index decrease |

Natural factors: Abundant resources can create better soil and climate conditions, so as to improve the basic ecological environment conditions of the region and provide favorable basic conditions for ecological restoration. If the ecological and environmental conditions of the initial study area are not good, or even bad, then we should talk about ecological restoration rather than ecological maintenance. Gui 'an has a good foundation of ecological environment quality. According to the calculation results of ecological quality in Gui 'an in this paper, 65.35% of the ecological remote sensing index (*RSEI*) in Gui 'an had an excellent grade in 2010, 61% of the landscape types were forest land, and the vegetation coverage was high. This provides favorable basic conditions for the recovery of ecological quality in the later period of construction, and also lays a solid foundation for the high-end, green and intensive development of Gui 'an. The ecological tourism industry is also the best choice for the sustainable development of new mountain towns. Compared with mountainous areas in Guizhou, where tourism is better developed, the development of Gui 'an involves large-scale land reclamation of low mountains and hills, while in mountainous areas in Guizhou, it is more based on the development and transformation of existing building foundations such as in Xijiang Qianhu Miao Village. In mountainous areas of Guizhou, even some towns developed on mountain land are mostly limited by natural factors, such as rock desertification. As a result, the development intensity and scale are insufficient, and the ecological system is easily damaged and difficult to recover. Therefore, it is more necessary to maintain the ecological quality level.

Factors for a planning strategy: The sustainable development of a region requires the government to make a correct development plan according to the actual local conditions, and the planning strategy of a region affects the protection and development of the region's ecological environment in the future. Lianjiang County government has made an ecological environmental protection plan for Gui 'an according to the three principles of sustainable development, zoning control, and the synchronization of development and environmental protection; the results of the calculation and analysis in this paper are consistent with the ecological environmental protection plan. In terms of time planning, Gui 'an takes 2016 as the intermediate node and develops and recovers quickly in accordance with the "two-stage" mode. According to the analysis results of the land-use change index and landscape pattern index, the change rate and change intensity of the first stage are larger than those of the second stage. Each stage includes two periods: development and expansion, and construction protection; each round of expansion is followed by the construction period, which pays attention to the construction and the maintenance of quality of the urban ecological environment while developing the economy. According to the results of ecological index calculation, the change in ecological quality is consistent with the land development model. During the first stage of development and expansion, the ecological quality decreased significantly, accounting for 45.64%. In the accompanying construction period, the control of ecological environmental damage was strengthened, and the ecological quality and vegetation coverage remained unchanged. During the second stage of development, expansion and construction, vegetation coverage and the ecological remote sensing index remained unchanged, while landscape structure and shape gradually stabilized. Moreover, ecological quality rebounded from 2016 to 2019, gradually approaching the middle level and becoming stable. Gui 'an effectively makes use of the local natural landscape pattern and reduces the role of inhibiting development due to topography and other natural conditions. This development mode greatly reduces the harm to the ecology brought about by the development of mountainous areas; this is an important reason to maintain ecological quality at a certain level while developing a large area in Gui 'an. In addition, according to the spatial distribution of land and the speed and intensity index of land change, the construction speed in Gui 'an is fast. The land development speed from 2010 to 2013 was more than eight times that of 2016 to 2019. By 2013, the urban land was basically developed in accordance with the demarcated urban development boundary. The land quickly changed from barren forest land to impervious urban land, with improved erosion resistance and geological stability, thus

realizing rapid terrain change, rapid restoration and rapid ecological restoration. According to the calculation results of the ecological remote sensing index, vegetation coverage index and landscape pattern index, both ecological quality and vegetation coverage changed greatly from 2010 to 2013, and remained unchanged from 2013; this is conducive to the restoration and improvement of regional ecological environmental quality in the later period. The degree of fragmentation of landscape types increased greatly in the first stage and decreased in the second stage. All of this indicates that rapid construction is of great significance to ecological maintenance in the process of mountain development. In terms of land-use planning, intensive use of urban space is an inevitable requirement for the sustainable use of urban land in the Xiashan area constrained by high ecological security [14]. It can be seen from the land classification results that the built-up land in Gui 'an is mainly concentrated in the Renshan area. According to the variation coefficient of the ecological remote sensing index and vegetation coverage, the two forest lands in the northeast and west directions of the Renshan area have always been undeveloped as ecological protection land, with a low fluctuation degree, protecting the natural landscape contour of the region to the maximum extent. Therefore, in the process of new mountain town development and construction, we can reduce the degree of ecological damage by choosing the right development mode and planning [59].

Market factors: Gui 'an is now a national 4A-level scenic spot. Fuzhou officially launched the establishment of the Gui 'an Xintiandi Leisure Tourism Resort as a national 5A-level scenic spot in 2016. It can be seen from the above results that Gui 'an was in the early stage of the second stage in 2016. From 2016 to 2021, the ecological remote sensing index and vegetation coverage remained unchanged, and the ecological quality rebounded. The area of ecological quality improvement increased from 18.14% to 27.94%, while the area of ecological quality decline only accounted for 16.48%. Since 2016, the changes in the regional landscape pattern have gradually decreased and have tended to be stable. Therefore, Gui 'an attaches great importance to the maintenance of related ecological issues in the process of applying for 5A scenic spots. Under such a market factor, Gui 'an will pay more attention to the protection of ecological land and increase investment in environmental governance, so as to bring more ecological benefits. At present, the land-use status of the region is stable. The boundary will not continue to expand, and this is more likely to expand the scope of influence to the adjacent areas of the township.

To sum up, we should pay more attention to ecological maintenance in the process of development and construction, instead of seeking a series of compensation after destroying the ecological environment. Although restoration is important, it cannot restore the ecosystem to its original state. Blindly seeking restoration without paying attention to protection in the process of development and construction will only lead to gradual deterioration of the ecological environment. As for the monitoring method of ecological maintenance, the ecological remote sensing index (*RSEI*) used in this paper is coupled with four indicators (greenness, dryness, humidity and heat), which can objectively, quickly and quantitatively evaluate the regional ecological environment, so as to rapidly monitor the regional ecological quality and distribution pattern. *FVC* is an index reflecting the change in surface vegetation. Better vegetation cover can maintain regional ecological balance. At the same time, *FVC* and *RSEI* were used to evaluate regional eco-environmental quality, and difference monitoring and coefficient of variation were used to monitor the changes and fluctuations in each period. The significant influence of new towns in mountainous areas on ecological environment quality after vegetation is replaced by built-up areas was revealed, and reflected the ecological environment quality and change in the study area. Through the comprehensive analysis of different landscape patterns and their evolution characteristics at multiple spatio-temporal scales, the corresponding change relationship among landscape pattern changes; land development and construction characteristics; and the ecological environment in the process of urban development in new mountainous areas was revealed; additionally, the mechanism of ecological maintenance was effectively explored. The research results provide effective solutions and references for how to realize

ecological maintenance in the process of development and construction. The importance of ecological maintenance can effectively make up for the deficiency of ecological restoration in protecting ecological environment quality and promoting sustainable development.

## 5. Conclusions

Regarding ecological maintenance, this paper finds that in the process of new mountain town development and construction, the degree of ecological damage can be reduced by selecting an appropriate development mode and making planning suitable for future development, according to the characteristics of the resources and the fragility of the ecological environment. Therefore, from the initial stage of development and construction to the stable development of this stage, the area maintains a certain level of ecological environmental quality. This is conducive to the protection of the region's ecological environment, but is also greatly conducive to the city's later ecological quality recovery. The use of land has a huge impact on the ecological environment of the region. The size of the impact and the spillover effect in the future must be considered in the development and construction process. We should not just consider the current benefits, nor should we pin our hopes on future continuous ecological restoration. Through this study, it was found that "two-stage" rapid development and the rapid restoration mode of land development in Gui 'an play an important role in the ecological maintenance of the region.

As for the research method of ecological maintenance, this paper used the optimized random forest classification and obtained a classification result with a kappa coefficient of 90% after field investigation and verification, by referring to a high-definition Google image; this is a relatively ideal method for accurately monitoring land-use change. The result is also in accordance with the land development principle of simultaneous development and environmental protection formulated by the Lianjiang County government to protect the environment. However, this paper did not compare the random forest classification results among different data, and in terms of feature selection, it is necessary to add feature elements such as elevation and slope in subsequent studies to further improve the framework of feature selection. With regard to ecological monitoring, through the analysis of the ecological remote sensing index, vegetation coverage index and the landscape pattern index, the comprehensive ecological quality of Gui 'an was effectively monitored. The results show that the ecological maintenance status is closely related to the land development pattern, and is also in accordance with objective facts. This indicates that the construction of such an evaluation system can comprehensively reflect the maintenance of ecological quality level in the process of regional development and construction to a certain extent. Secondly, in addition to human activities, some special natural factors, such as natural fire, will also cause the decline of regional ecological environment quality. Therefore, it is necessary to consider whether there is natural interference when conducting research on regional ecological maintenance. This will also increase the complexity of the methods used to analyze the problem of ecological maintenance. At present, there is no complete method system for the study of ecological maintenance. In the future, further systematic research and exploration should be carried out on the basic concepts, technical methods and engineering measures of ecological maintenance, so as to comprehensively build the theory and methodology of ecological maintenance in different regions.

**Author Contributions:** Conceptualization, R.H. and J.S.; data curation, R.H. and J.S.; methodology, R.H., J.S. and X.L.; validation, R.H., J.S., X.L. and Z.L.; formal analysis, R.H. and J.S.; investigation, R.H.; resources, R.H., J.S. and X.L.; writing—original draft preparation, R.H.; writing—review and editing, R.H., J.S., X.L., Z.L. and S.L.; visualization, R.H.; supervision, J.S., X.L. and R.H.; project administration, J.S., X.L., S.L., Z.L., Q.L. and J.W.; software, R.H.; funding acquisition, J.S., X.L., S.L. and Z.L. All authors have read and agreed to the published version of the manuscript.

**Funding:** This research was funded by the Multigovernment International Science and Technology Innovation Cooperation Key Project of the National Key Research and Development Program of China (grant number 2018YFE0184300), and "GIS and Remote Sensing for Sustainable Forestry and

Ecology (SUFOGIS)"(598838-EPP-1-2018-EL-EPPKA2-CBHE-JP) supported by EU Erasmus + Project, October 2018–October 2022.

**Data Availability Statement:** Not applicable.

**Acknowledgments:** The author thanks the organizations that provided the original data and software: the China Geospatial Data Cloud Platform (http://www.gscloud.cn/ (accessed on 12 July 2021)); the European Space Agency (https://scihub.conpernicus.eu/dhus/#home (accessed on 13 July 2021)); GlobeLand30 (http://www.globallandcover.com/ (accessed on 30 June 2021)); the China Geographic Information Resources Directory Service System (https://www.webmap.cn/ (accessed on 18 September 2021)); Sen2Cor software (https://scihub.copernicus.eu/dhus/#/home (accessed on 11 July 2021)); and SNAP software (http://step.esa.int/main/download/snap-download/ (accessed on 13 July 2021)).

**Conflicts of Interest:** The author declares no conflict of interests.

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
