# Peer review of "Remote Sensing Analysis of Ecological Maintenance in Subtropical Coastal Mountain Area, China"

_remotesensing, doi:10.3390/rs14122734_

Round 1

Reviewer 1 Report

1, 4 to 18, 20, 22. Ok

2, 3, 19. Information may be interesting to add (if not)

21: use “urban ecosystem” instead of “urban ecology” which may refers to the science studying the ecosystems?

Reviewer 2 Report

An interesting work that is methodically applicable in other regions of the world.

- What the authors consider Mountain areas (altitude specification)

- There is a lack of supporting literature in ecosystem services, I recommend resp. must be added to the bibliography of authors De Groot, Burkhard, B ......

- It would be appropriate to add a position map of the surveyed area within China.

- Figure 11 and 14 - it is necessary to add that the white color (water) is unchanged

- The authors use the term ecosystem services in the introduction, but it is no longer in the discussion. How to use the results of the work in the evaluation of ecosystem services?

Reviewer 3 Report

Title - can be simplified; the word exemplary is not necessary to be used (for example use a case study from Gui an Mountains, China)

Abstract - the text is too long and can create difficulties in understanding the scope and the objectives but especially the significance of the results. It is essential to provide key information regarding data, data processing and analysis, results obtained and their significance, including validation. This order need to be followed for the text structure.

Introduction

Ecological terminology employed must be from scientifical literature. Please check again. Usually avoid general statements. It is necessary to adapt the instroduction to the scope of the journal and put the emphasis on the role of remote sensing in ecological services analysis and management and especially on mountain areas related approaches.

Poor references in remote sensing related subject are displayed. 

Extract more clear the objectives of the approach.  It is difficult to identify them, to the end of a general and descriptive introduction section, not so much adapted to the current approach.

Materials and method

Study area can be better explained. It is useful to argue its selection in the context of the approach and to provide only the most significant geographic features for understanding of its significance (partly done but not rigurous). Geographical names are not mapped every time and the reader need to search them on other maps. Figure one is of poor quality and not easy to read. Mapped area need to be located in a simple National map and after showed in a simple 2d and 3d models (hypsometric map has no geographica names and colours are not good, and 3d model is little and not explained - provide peaks and few settlement at a bigger display scale).

Data sources - are only described and less explained (plus and minus). Table 1 is not complete (ex. WRS system path/row is only for Landsat data and not for Sentinel-2 MSI data). Other data like DEMs need to be included in the table. 

The flow chart in figure 2 is essential in understanding the approach and need much more explanation (please argue your choices in image processing and new data production for practical usage in ecological maintenance).

Tables 2 and 3 are useful but before you need to explain and not to describe the performances of data calibration. As we know in rugged montane terrain there are strong topographical influences, including shadow and illumimation. How to manage these differences in data normalization. Is it possible to provide some examples from data normalization process?

Section 2.2.2. to 2.2.8 are theoretical presentations of various indices employed. Mathematical formulae are only described and not fully adapted to the current approach. It is difficult to red it and references are enough to guide the reader to details. Extract much better the significance for the mountain area ecological change features mapping. Simplify the presentation... You can use a table to provide this data in a synthetic overview.

There are references that can be partly provided to the state-of-the art section of the paper.

Results

Maps in figure 4 are general and difficult to red. Display them properly and/or focus on the most transformed areas identified.

Figure 8 can be better displayed in order to understand it.

Section 3.2 is an interpretation of resulst and could be a part of the discussion part of the manuscript.

Figures 9,10, 11, 13, 14 (maps) can be better adapted to be red. Using a basic key map (of study area) with geographical names and some details focused on most transformed areas, you can obtain more expressive representation (a recommendation).

Figure 9 - lower is correct (legend) - same in table 6.

It might be easier to explain in a table the features of each classes of the proxies used in analysis. Text is difficult to be followed. A systhematic presentation of results need to be done from simplest to the most complex models. Avoid details of strictly local importance as well as redundant explanation too.

Section 3.3 can be better included in discussion section but with connecting the results with the ecological and landscape planning strategies in order to extract the practical significance of the approach. Avoid simple geographical description. Focus on the remote sensing approach and its validation on the ground.

Discussion section must include the very interpretation of results (most of these are in previous section) and less general explanation. Kappa coefficients of different models obtained can be of a real value in order to extract the most significant transformation to be integrated with decision making approaches.

If possible, you need to compare your results and their significance with similar ones from China mountain regions and not only. After the complexity of methodology can be disussed accordingly, in order to extract the most significant data layers and models.

By adapting statistical data produced to the ground truth and local/regional ecological planning issues it is easier to extract the practical use of the analysis (really important in sustainable development of this type of regions).

Conclusion - must be adapted to the objectives and results than to the regional planning issues and afterwards to the opening of them to the practical needs. 

Round 2

Reviewer 3 Report

Title - What about the meaning of the exemplary word in the title?

Introduction - is better focused on the approach current issues than the first version but avoid general statements. Remote sensing in mountain built-up area is the topic to be documented and explained.

Objectives must be more visible to the end of the introduction.

How is better to write- mountain towns or mountain cities? 

Figure 1 - legend it is not correct to use DEM - write elevation (m) as you derived this data from the model. It is better to make transparent the yellow polygons from the maps to the right. 

Check please if all geographical names in the text occurs on the maps.

Row 196 - nearest neighbor is a method or an algorithm?

Tables 2 and 3 - indices and other features can be accompanied by references as they are from literature.

Figures 6 and 7 - rate of change and change intensity diagrams are really useful and can help the reader but I do not see the explanation of the indices of change you employed.

Fig. 8. - the name can be modified as contrast word could be confusing (use for example - recent and fast (?) urban landscape transformation sample (specify the situation - building of...). 

Discussion - usually needs an explanation of the role of methodology and secondary of the regional significance of results. If possible you need to calibrate your methodology (qualitative, even quantitative) to similar approaches and explain the added value of your approach (it really exists and is partly explained).

It is an interesting approach with an innovative component with a real practical value.

Some terrain images (before and after orthoimagery for example) can be helpful, too.

Supplementary material is good but I do not consider you need to provide data about image or product resolutions. Only a reference in the text can help. Indices employed related references are fitting here in a table (an idea).

Author Response

This manuscript is a resubmission of an earlier submission. The following is a list of the peer review reports and author responses from that submission.

Round 1

Reviewer 1 Report

Dear authors,

I enjoyed reading your work on a relevant topic. You will find my comments enclosed in the joint file. Best regards

Reviewer 2 Report

The paper describes a study of the changes in land use land cover in a touristic mountainous area. The context and concepts should be more precisely defined. Results about land changes are interesting. The drivers are described in the second part of the article: why this choice? Is it part of the research and findings? It could be integrated into a larger discussion section.

Here some comments about precise part of the article:

Abstract

-Explicit the « two-stage » pattern: stage in time, between 2010 and 2021?

  1. Introduction

-Distribution of the mountainous areas in China: inland only?

-To define more clearly: ecological maintenance. The notion is defined by its means. How different is it to monitoring and protection. Is it services?

-The end of the paragraph with presentation of the steps of the methods and the approach could be reorganized to facilitate reading. Is it a description of the methodological steps (indexes, analyze of development conditions…)?

  1. Materials and Methods

-To be presented in the introduction the fact that the study site is a Hot Spring tourist place?

How representative is the area compared to the all mountain area in China?

-References or website to add for the existing tools used to process the data (Sen2Cor…)

-I did not see the quotes for tables and figures in the text. For instance, description of the Figure 3 quoting the main steps in the text?

- « Optimization of random forest classification” is a aim of the research or a mean?

-Spectral bands to describe, even not in detail? Corresponding to the wavelength…

-In the text and table, two spellings:  greenness and greeness

-Table 2 and 3, Figures 4 and 5, to be described in the text. Tables and figures should support the description of the defined Landsat Feature and closely related to the text.

-“4 evaluation indexes, such as” -> rather “corresponding to”?

  1. Results

- Figure 6: missing dates for two maps

-Why tables 6, 7, 8 take into account the years 2013 and 2019 (already mentioned in the abstract) whereas there is two stages in the process (before and after 2016)?

-Explanations about the process of changes from forest to bare land then to built-up areas: how long is it planned in advance? How long does it stay bare land in average (a year, 6 months, less)? Is there communication about it? Related land planning and policies?

-Interesting description of the results. “ecological maintenance and improvement »: does it mean improvement and increasing of ecological services?

  1. Diving factors analyis

-Figure 16: to be explicit, digital elevation model, units in meters…

-Lots of consequences of the hot springs infrastructures: heath care, services, etc. How many inhabitants and tourist? Which dynamics of the tourism and length of stays (one day, a week, more)?

  1. Discussion

-“guian » : Gui’an ?

-“urban ecology”: meaning? “ecological construction »

-Discussion about the drivers?

Reviewer 3 Report

Line 12 "Mountainous areas" - must be specified from what altitude are Mountainous areas
In the Figure 2, the study area line definition is missing
In the Figure 6, not all parts of the year are indicated
In the Figure 10 it would be appropriate to add 1 - worst, 5 - best to the legend
It would be appropriate to add In Figure 12 and 15, an area that has not changed land use throughout the period under review
Figure 16 move after text (line 517)
It would be appropriate to add to the conclusion how other planned (approved) activities will affect the change of current land use

Reviewer 4 Report

Abstract

rows 19-27 the sentence is long and difficult to follow. Rewrite it please.

Usually it is recommended to focus on the  key objectives and the mehodological findings of the manuscript. 

Introduction - the presentation is good and useful, but there is a need of integration of international references regardind these issues. There are several papers in different international journals on this subject like fox example Mountain Research and Development, Revue de Geographie Alpine and even Land Use Policy or Landscape and Urban Planning. 

The role of remote sensing approaches on these subjects must be emphasized. You can also extract some achievements and focus on the less analyzed issues. I

Objectives of the current approach must be visible and presented to the end of the introductory section. Please follow the idea of a methodological approach than of a regional applied geographical approach. It is an interesting example to integrate remote sensing analysis to a regional planning issue in mountain area sustainable management context.

Section 2.2 in methodology chapter is of course the core of the approach. It is interesting for a lot of potential readers to discover new possibilities in optical image classification. This is the reason it might be better to focus a little on the argued explanation of the selection of the group of layers, including derived texture and indices. Some examples can be helpful.

Sub-sections 2.2.2 - 2.2.7 are important but it might be better to focus on the current problem of the aprroach and not on simple theorical aspects (adapt theory to the case analysis, extracting significant methodological issues). References are also important when defining them, if there were employed different sources. Abbreviations must be carefully explained.

All transformation explained in this section must be integrated in the flowchart scheme of the approach in figure 3 (check if this is adequately solved).

Avoid general statements and adapt the section to the special features of the current study.

Results 

Figure 6 is interesting. and need a description of classes together with the ecological significance in the planning context of the local/regional development  (here, very general) If possible it might be better to be displayed at more detailed scales. Some geographical names used in the text can occur on the maps (ex. Renshan district and other). Some field photos from above or aerial images samples on same sites (old and new),  could be interesting.

Figures 10, 11 and 12 are important and need a more adequated explanation of the mapped areas significance. 

Paper has enough results and I suggest to integrate their presentation with text sections.

Maps in figures 10-14 are rather general and it is a need of detail mapping of the representative areas according to the explanation of results. Focus on two areas with opposite trends in ecological transformation in order to understand the results.

4. Driving factors analysis is descriptive and not well connected to the theme of the paper. It can be better adapted by integrating spatial and temporal changes with the regional context issues in a graphic and map formulae. For example, figures 16 and 17 can be preserved but with superposing layers derived from classification, starting with the built-up areas and their transformation. In figure 16 we cannot understand the topographical context as the legend is by default and not suitable. Usually altitude and slope gradients can be a limitation, but there is also a need to explain the slope morphodynamics like slides or streams leading to some ecological features of the tourist settlement development.

Usually the mentioned geographical names need to be mapped.  

What about the tourist resource sustainable use and its spatial and temporal features? - partly described

Transportation routes and facilities can be also mapped in order to understand them as driving factor, by adapting them to figure 17.

Do not miss that the driving factor issues must be in a good continuity with the current approach focused on remote sensing and ecological mapping of landscape transformation. Find a smooth transition between these aspects.

Some elements can be extracted to be integrated in regional planning strategies, as well.

Discussion - is not adequately provided.

This section is in fact the interpretation of the results and the explanation of the methodological positive and negative features.

Extract the significance of your findings and try to calibrate the results with similar results produced from similar remote sensing data in similar approaches from international literature.

The paper need to be opened to methodological issues and less to the regional ecological study of landscape change. It is necessary to find similar approaches.

What is important -  the comparison of kappa coefficients of multitemporal image classification and to validate the indices with field situation and even with planning data or other independent data.

Conclusion - could be an adaptation of the results and of the methodology to the future potential regional planning sustainable development strategies. Follow the objectives defined to the beginning and evaluate the degree you touched them (explain the limitations after the positive achievements).